# Sample-to-answer platform for the clinical evaluation of COVID-19 using a deep learning-assisted smartphone-based assay

Seungmin Lee[1,2,9], Sunmok Kim[1,9], Dae Sung Yoon [2,3,4,9], Jeong Soo Park[1], Hyowon Woo[1], Dongho Lee[5], Sung-Yeon Cho[6,7], Chulmin Park [6], Yong Kyoung Yoo[8] ✉, Ki-Baek Lee[1] ✉ & Jeong Hoon Lee [1] ✉

Since many lateral flow assays (LFA) are tested daily, the improvement in accuracy can greatly impact individual patient care and public health. However, current self-testing for COVID-19 detection suffers from low accuracy, mainly due to the LFA sensitivity and reading ambiguities. Here, we present deep learning-assisted smartphone-based LFA (SMART[AI]-LFA) diagnostics to provide accurate decisions with higher sensitivity. Combining clinical data learning and two-step algorithms enables a cradle-free on-site assay with higher accuracy than the untrained individuals and human experts via blind tests of clinical data ($n = 1500$). We acquired 98% accuracy across 135 smartphone application-based clinical tests with different users/smartphones. Furthermore, with more low-titer tests, we observed that the accuracy of SMART[AI]-LFA was maintained at over 99% while there was a significant decrease in human accuracy, indicating the reliable performance of SMART[AI]-LFA. We envision a smartphone-based SMART[AI]-LFA that allows continuously enhanced performance by adding clinical tests and satisfies the new criterion for digitalized real-time diagnostics.

Recent diagnostic strategies for handling pandemics suggest that frequent, inexpensive, simple, and rapid tests can help minimize the spread of the severe acute respiratory syndrome coronavirus 2 (SARS-CoV-2) virus[1,2]. Although the real-time polymerase chain reaction (RT-qPCR) test is highly sensitive, frequent on-site tests of COVID-19 using RT-qPCR are still challenging. Generally, RT-qPCR can detect viral shedding up to 17 days after the infection period, increasing unnecessary quarantine[2,3]. Moreover, long turnaround times of RT-qPCR allow infected people to spread the viruses exponentially before getting results.

Lateral flow assays (LFAs) are disposable, fast, inexpensive, convenient, and easy to use, and therefore, they are the best candidates in point-of-care testing (POCT) for clinical diagnosis to develop a fast and frequent test for COVID-19[2,4]. However, LFAs lack reliability and accuracy compared to traditional laboratory assays; moreover, untrained individuals analyze most LFAs with naked-eye detection, which inevitably limits the accuracy, especially for low virus titers[5–7]. To achieve high sensitivity and accuracy, hardware-based approaches such as assay optimization (reagent and receptor)[8–10], signal enhancement[11,12], sample enrichment[13,14], and signal amplification[15–17] have been reported.

[1]Department of Electrical Engineering, Kwangwoon University, 20 Kwangwoon-ro, Nowon, Seoul 01897, Republic of Korea. [2]School of Biomedical Engineering, Korea University, 145 Anam-ro, Seongbuk, Seoul 02841, Republic of Korea. [3]Interdisciplinary Program in Precision Public Health, Korea University, Seoul 02841, Republic of Korea. [4]Astrion Inc, Seoul 02841, Republic of Korea. [5]CALTH Inc., Changeop-ro 54, Seongnam, Gyeonggi 13449, Republic of Korea. [6]Vaccine Bio Research Institute, College of Medicine, The Catholic University of Korea, Seoul, Republic of Korea. [7]Division of Infectious Diseases, Department of Internal Medicine, College of Medicine, The Catholic University of Korea, Seoul, Republic of Korea. [8]Department of Electronic Engineering, Catholic Kwandong University, 24, Beomil-ro 579 beon-gil, Gangneung-si, Gangwon-do 25601, Republic of Korea. [9]These authors contributed equally: Seungmin Lee, Sunmok Kim, Dae Sung Yoon. ✉e-mail: yongkyoung0108@cku.ac.kr; kblee@kw.ac.kr; jhlee@kw.ac.kr

Artificial intelligence (AI)-assisted approaches for biomedical applications can improve the reliability and accuracy of sensors such that they are comparable to human performances[18–21]. Smartphone-based diagnostics can be considered a potential candidate for POCT because it is easy to acquire digitalized images from a smartphone for user-friendly diagnostics, enabling home- and self-tests with digital connectivity[22–26]. Since the number of current smartphone users exceeds 6 billion, meaning >80% of the world's population owns smartphones (https://www.statista.com/statistics/330695/number-of-smartphone-users-worldwide/), the assay using a smartphone can give accessibility and affordability. Smartphone-based assays have been performed to detect sperm concentration and motility[27], protein biomarkers[28], CRISPR-read SARS-CoV-2[25], Zika virus[29], norovirus[30], cell migration[31], SARS-CoV-2 variants at single-nucleotide resolution[32]. To further improve the performance of smartphone assays, AI-assisted assays were performed for DNA diagnosis in malaria detection[24], HIV rapid tests[33], CRISPR-Cas13a based SARS-CoV-2 detection[26], and serological SARS-CoV-2 antibody test[21].

Here, we present deep learning-assisted smartphone-based LFA (SMART[AI]-LFA). We design a two-step CNN model (object finding and classification) that can efficiently and accurately provide output results. This makes it more suitable for use in various environments without external cradles. We train the algorithm with SARS-CoV-2 nucleocapsid protein spiked into phosphate buffered saline (PBS) buffer (referred as standard data) ($n = 8914$), retrain it with additional clinical data ($n = 8005$) to strengthen the model, and acquire excellent specificity and accuracy for testing clinical data. Moreover, we discuss a blind test of untrained individuals, human experts, and SMART[AI]-LFA that demonstrate high levels of sensitivity and specificity of SMART[AI]-LFA; this shows the feasibility of the cradle-free sample-to-answer platform with digitalized real-time connectivity.

## Results
### Workflow and experimental design
Figure 1 shows the workflow of the SMART[AI]-LFA and experimental design of the entire study that enables a sample-to-answer platform for COVID-19 with the aid of deep learning-assisted determination. The most significant advantage of SMART[AI]-LFA is that it requires no external cradles attached to the smartphone. Since each smartphone has a different size and camera location, the one-cradle-fits-all approach may not work. Moreover, many external cradles need their external optics[27,31,32,34], fluorescence components[25], microscope[28,30], and Bluetooth[24,35] for reading and transmitting signals to smartphones, consequently increasing costs. Meanwhile, the use of AI could eliminate external cradles, which meets the universality of smartphone-based assays discussed later. Further, a smartphone-based AI provides a great opportunity to meet the REASSURED criteria (Real-time connectivity, Ease of specimen collection, Affordable, Sensitive, Specific, User friendly, Rapid and robust, Equipment free, and Deliverable to end-users)[22], which are the new criteria for digital connectivity. We develop a deep learning algorithm that works with a smartphone application to provide results (positive/negative/invalid) of the SARS-CoV-2 antigen test; this helps reduce reading ambiguities.

The architecture of the SMART[AI]-LFA includes an algorithm for performing object finding, which includes cropping the region of interest (ROI) from the entire LFA, and another for classification from colorimetric intensity (Fig. 1a). To this end, we train a dataset using three different smartphones (two Android and one iOS) for the best object finding. Further, the classification algorithm is used that can suggest a decision model. We additionally used an augmented dataset from the original one for standard data ($n = 8914$) because insufficient data leads to poor accuracy and reliability of the deep learning methods. After qualifying the clinical samples, we trained the model with additional clinical data ($n = 8005$). Then, we evaluated the predictive power of COVID-19 using our SMART[AI]-LFA architecture. We

first designed an app to pair with AllCheck COVID19 Ag (Calth Inc.), then tested the seven commercial COVID-19 models such as Panbio COVID-19 Ag (Abbott), BIO CREDIT COVID-19 Ag (Rapigen), SGT-flex COVID-19 Ag (Sugentech), GENEDIA COVID-19 (GCMS), COVID-19 Ag Test (Humasis), COVID-19 Ag (Genbody), and InstaView COVID-19 (SGmedical). The image of the test results acquired using a smartphone was sent to the server where the two algorithms are located (Amazon Web Services, AWS); then, the test results were sent to individual users.

We determined the accuracy of the blind test using 1,500 test images from untrained individuals ($n = 10$), human experts ($n = 10$), and SMART[AI]-LFA (Fig. 1b). Clinical samples ($n = 65$, COVID-19 patients: 45 and healthy controls: 20) were tested to validate the clinical predictability of SMART[AI]-LFA. We arranged the training, test and validation datasets for algorithm development (Supplementary Tables 1–2, Supplementary Fig. 1).

### Model optimization for object finding and classification
To satisfy the sample-to-answer diagnostic strategy, we developed AI algorithms consisting of object finding and classification (Fig. 2). Many existing methods do not work well under different surroundings without external cradles because diagnostic surroundings include cases such as indoors/outdoors, lighting conditions, and shade/sunlight[25,36]. Further, color temperature and background images can severely affect the accuracy of image-based analysis[37,38]. Therefore, we developed two algorithms to enhance the performance of the algorithm: one for finding objects and the other for classification from colorimetric intensity (Fig. 2a). Our SMART[AI]-LFA was evaluated under the AWS environment using deep learning frameworks (YOLOv3 for object finding and ResNet-18 for classification) that return the assay results to the smartphone applications.

We obtained the receiver operating characteristic (ROC) curve and validated the prediction accuracy of object finding using three different approaches with 8914 standard training data (positive: 5801 and negative: 3113) and 1458 clinical test data (positive: 1026 and negative: 432) (Fig. 2b, c). Then, we selected objects to achieve better accuracy through the deep learning process. The first approach detects the entire LFA cassette used by previous studies[21] (Cassette all), the second detects the LFA cassette window where the test/control line is included (Window), and the third detects only the test line (Test line (TL) only). We used YOLOv3 for automatic object selection. Currently, object detection is performed using color-based detection[39] and contour-based detection[40]; however, these two detection methods exhibit limited performance. Color-based detection, which detects the red color in the LFA kit, inevitably limits the performance for low test line signals. Contour-based detection is affected by the background color or patterns of the image (Supplementary Fig. 2). Mendels et al.[21] reported an AI-based SARS-CoV-2 antibody (Ab) serological test using the training of the entire LFA cassette (shown as cassette all in Fig. 2b, c). Interestingly, the prediction accuracy using training of the cassette all (yellow line in Fig. 2b) was 74.3%. However, with window (blue line in Fig. 2b), we increased the accuracy to 87.1%. Furthermore, we maximized the accuracy by 94.8% when using test line only for the ROI (red line Fig. 2b), indicating the high dependency on ROI for prediction accuracy (Fig. 2b, c). We believe that the best performance of test line only can be attributed to the meaningful information in the original image being concentrated in a very small area near the test line.

The deep learning network can identify the classification shown in Fig. 2d, e. We prepared a training dataset (positive: 5801; negative: 3113) and a test dataset (positive: 1026; negative: 432) with random shuffle. Seven frameworks (DenseNet-121[41], DenseNet-161[41], ResNet-18[42], ResNet-34[42], ResNet-50[42], MobileNetV2[43], and SqueezeNet[44]) were used, and the root mean square error (RMSE) values of these seven models confirmed ResNet-18 and 50 as the best models for classification (Fig. 2d). The numbers '18' and '50' refer to the number of layers,

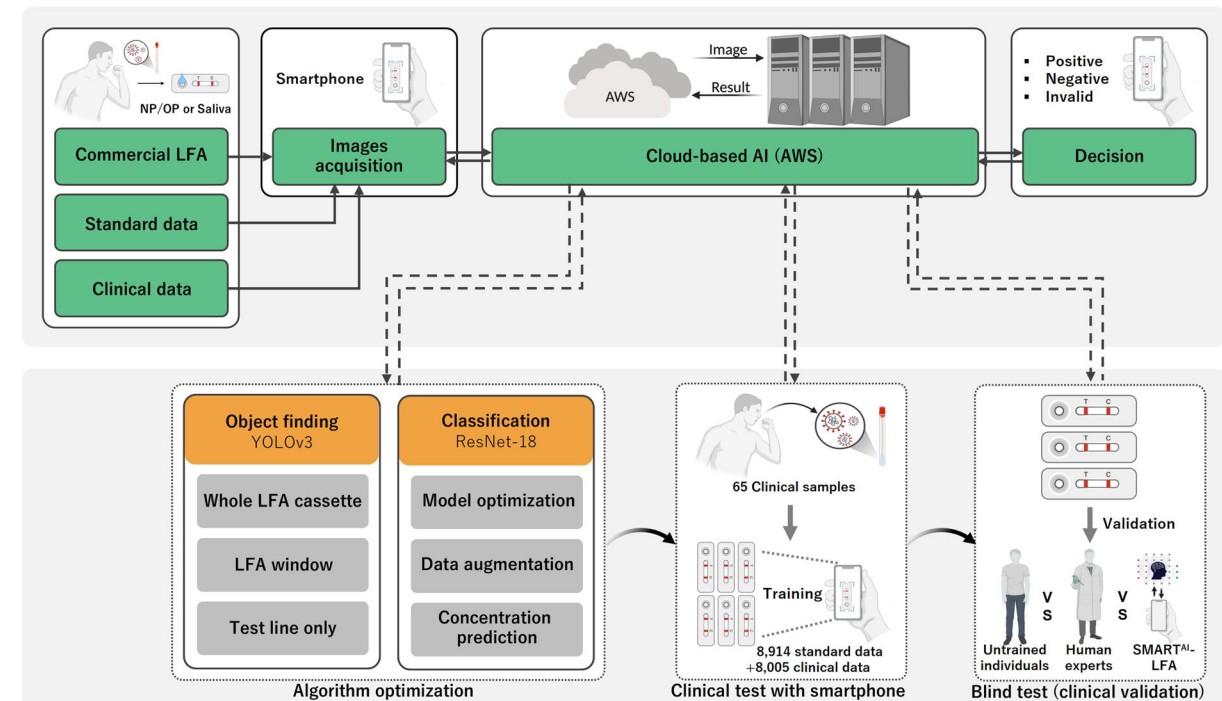

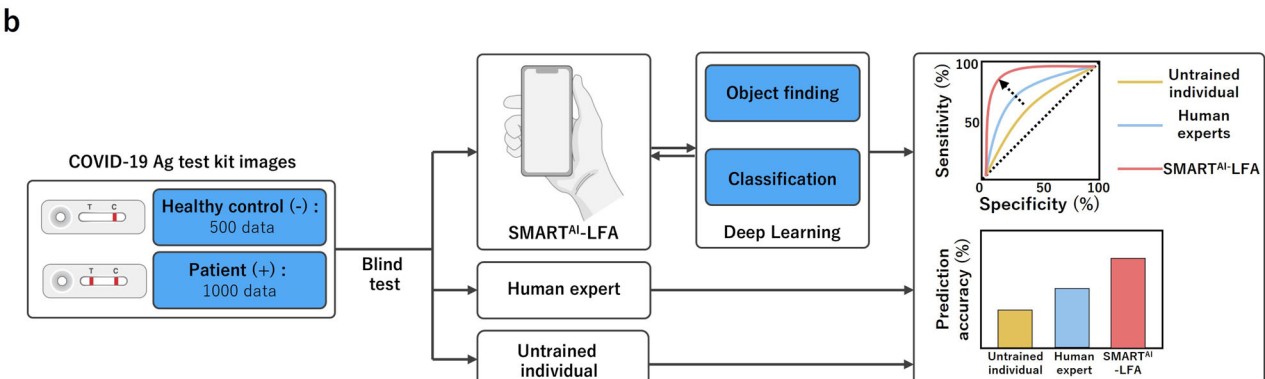

**Fig. 1 | Workflow of SMART^AI-LFA for COVID-19 diagnostics. a** Experimental design of SMART^AI-LFA, enabling sample-to-answer for COVID-19 with the aid of deep learning-assisted determination. The architecture mainly consists of object finding and binary classification, which are trained over the dataset containing 16,919 COVID-19 tests (11,468 positive and 5451 negatives). **b** The main validation of SMART^AI-LFA by the blind test (*n* = 1500) from untrained individuals, human experts, and SMART^AI-LFA, validating SMART^AI-LFA's clinical predictability. SMAR-T^AI-LFA deep learning-assisted smartphone-based LFA, LFA lateral flow assay.

and the model with 50 layers (ResNet-50) performed better on the basis of RMSE. However, more layers require more computing time to determine the results; therefore, we chose ResNet-18 model, enabling a highly accurate diagnosis.

### Data augmentation for improving performances

Data augmentation is a powerful tool for improving the performance of deep learning models. Deep learning models demand a large dataset for training. However, increasing the number of training data has physical limitations, such as time consumption, human resource, and cost. The number of training data can be increased by reproducing already existing data (data augmentation). Multiple reproduced data can be obtained by applying various combinations of crops and resizing them to a single image. We conducted data augmentation as shown in Fig. 2f; the two main parameters for data augmentation are color temperature and brightness modulation. To avoid bias due to data augmentation, we prepared images under various surroundings;

in turn, we could increase the accuracy with data augmentation (Fig. 2e, f). We prepared five datasets to train SMART^AI-LFA and showed the prediction accuracy (Fig. 2e). We first acquired data from three smartphones with one lighting condition (single light (SL), gray bar), and data augmentation was performed (augmented SL, blue bar). We then added multiple lighting conditions (multi light (ML), yellow bar) followed by data augmentation (augmented ML, purple bar). Finally, we add indoor/outdoor light conditions, day/night, various backgrounds, and shadows, and conducted data augmentation (all surroundings, red bar). We validated the enhanced prediction accuracy from 60 to 89% with the first data augmentation of one light condition; adding multiple lighting conditions and data augmentation increased the accuracy from 85 to 90%. Finally, adding all the considered surroundings, i.e., indoor/outdoor, day/night, various backgrounds, and shadows, helped enhance the accuracy from 90 to 99.2%. This indicated that the acquisition of data and its augmentation helps enhance the accuracy of the deep learning-based smartphone assay.

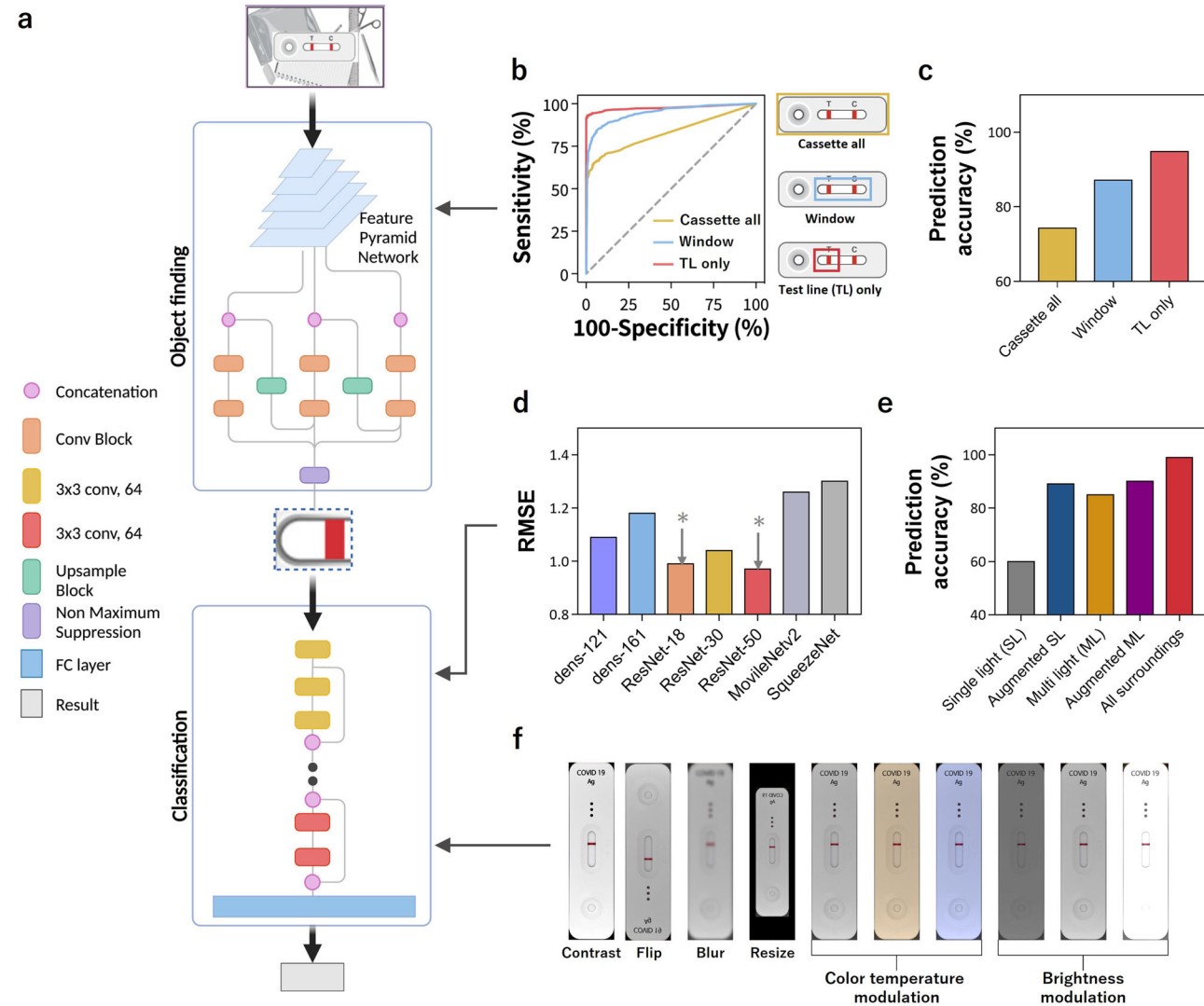

**Fig. 2 | Algorithm optimization. a** The algorithm of SMART[AI]-LFA consists of object findings and classification. **b**, **c** Algorithm #1: Object finding; **b** ROC curves, and **c** prediction accuracy of object findings using three different approaches. Detection of only the test line reveals a higher accuracy than the whole LFA cassette and LFA cassette window. **d**–**f** Algorithm #2: Classification; **d** RMSE values from seven different frameworks, showing ResNet-18 and 50, provide a high-accuracy diagnosis. **e** Training of SMART[AI]-LFA; we prepared five data sets and represented acquiring data and its data augmentation, allowing enhanced accuracy of deep learning-based smartphone assay. **f** Examples of data augmentations. SMART[AI]-LFA deep learning-assisted smartphone-based LFA, ROC receiver operating characteristic, LFA lateral flow assay, RMSE root mean square error.

## Blind tests using clinical samples

Figure 3a illustrates the process of clinical evaluation via blind tests. We assess the blind test using three different groups: untrained individuals, human experts, and SMART[AI]-LFA using 1500 test images (1000 positives and 500 negatives). We collected clinical samples from COVID-19 patients at Seoul St. Mary's Hospital. The information pertained to SARS-CoV-2 patients ($n = 45$) and healthy controls ($n = 20$) including sample collection, variants, sex, ages, and Ct values (Supplementary Table 3). All the samples were analyzed with RT-qPCR, then conducted the LFA assay. Finally, the positive data from LFA assay were classified into four groups, i.e., high/middle/middle-low/low titer, with the aid of the color chart level (high with levels 10–8, middle with level 7–5, middle-low with level 4–3, and low titer with level 2–1 for positive, and negative with level 0). For positive data ($n = 1000$), we evenly distributed data across four groups ($n = 250$). We prepared negative data (health controls, $n = 500$). For the blind test, ten untrained individuals and ten human experts tested each of the 150 test images, which included 25 high, 25 middle, 25 middle-low, 25 low, and 50 negative data. The sum of the

blind tests for both untrained individuals and human experts was 1500 images.

The ROC curve shows a general overview of the three different models; a larger value of the area under the curve (AUC) indicates a better classifier. From the ROC curve in Fig. 3b, we observe larger AUCs for SMART[AI]-LFA (1.00) compared with that for untrained individuals (0.79) and human experts (0.86); this demonstrates that SMART[AI]-LFA is an excellent classifier for clinical assays. Figure 3c shows the table for the three different groups, which indicates a considerable enhancement in sensitivity and specificity using a SMART[AI]-LFA (100 and 100%) compared with untrained individuals (72.9 and 86.0%) and human experts (83 and 88.2%). To closely study the reason for the accuracy reaching 100%, we explored the effect of the training and test datasets discussed later (Fig. 4g, h).

We present three positive clinical sample images in Fig. 3d to clarify the AI's decision ability. Figure 3e shows the evidence of a positive test line of Fig. 3d with contrast enhancement. Although all groups can provide the correct answer as seen from the first image, only SMART[AI]-LFA can predict the positive samples from the third

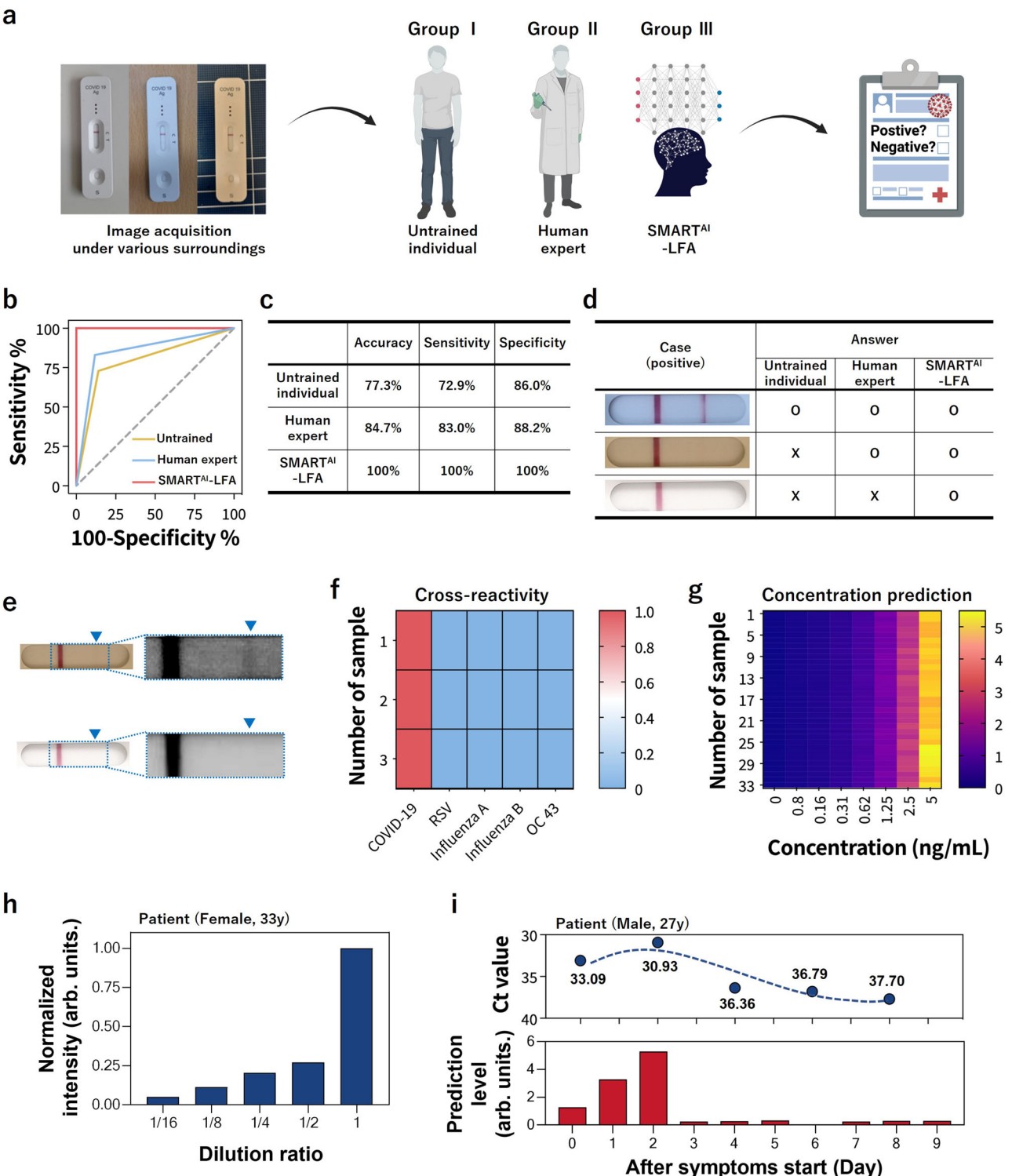

**Fig. 3 | Clinical validation via blind tests, cross-reactivity, concentration prediction, and daily monitoring. a–e** Blind test; **a** Workflow of the clinical validation process using a blind test using 1500 test images ($n = 1000$ patients and $n = 500$ healthy controls). Using SMART[AI]-LFA, we carried out blind tests ($n = 1500$) of untrained individuals ($n = 10$) and human experts ($n = 10$) every 150 test images and compared it with SMART[AI]-LFA results, showing great enhancement in sensitivity, specificity, and accuracy using a SMART[AI]-LFA. **b** The ROC curve and **c** prediction accuracy. **d**, **e** Answers for three positive clinical sample images, clarifying the AI's decision ability. **f** Cross-reactivity using different respiratory viruses, revealing no cross-reactivity. **g** The concentration prediction ability of SMART[AI]-LFA using a heat map, representing the ability of quantitative analysis. **h** The sample concentration prediction with clinical patient sample (female, 33 y) according to dilution factors. **i** Daily COVID-19 test of clinical sample (Male, 27 y), showing the ability of daily monitoring of virus titers via SMART[AI]-LFA. SMART[AI]-LFA deep learning-assisted smartphone-based LFA, ROC receiver operating characteristic, AI artificial intelligence.

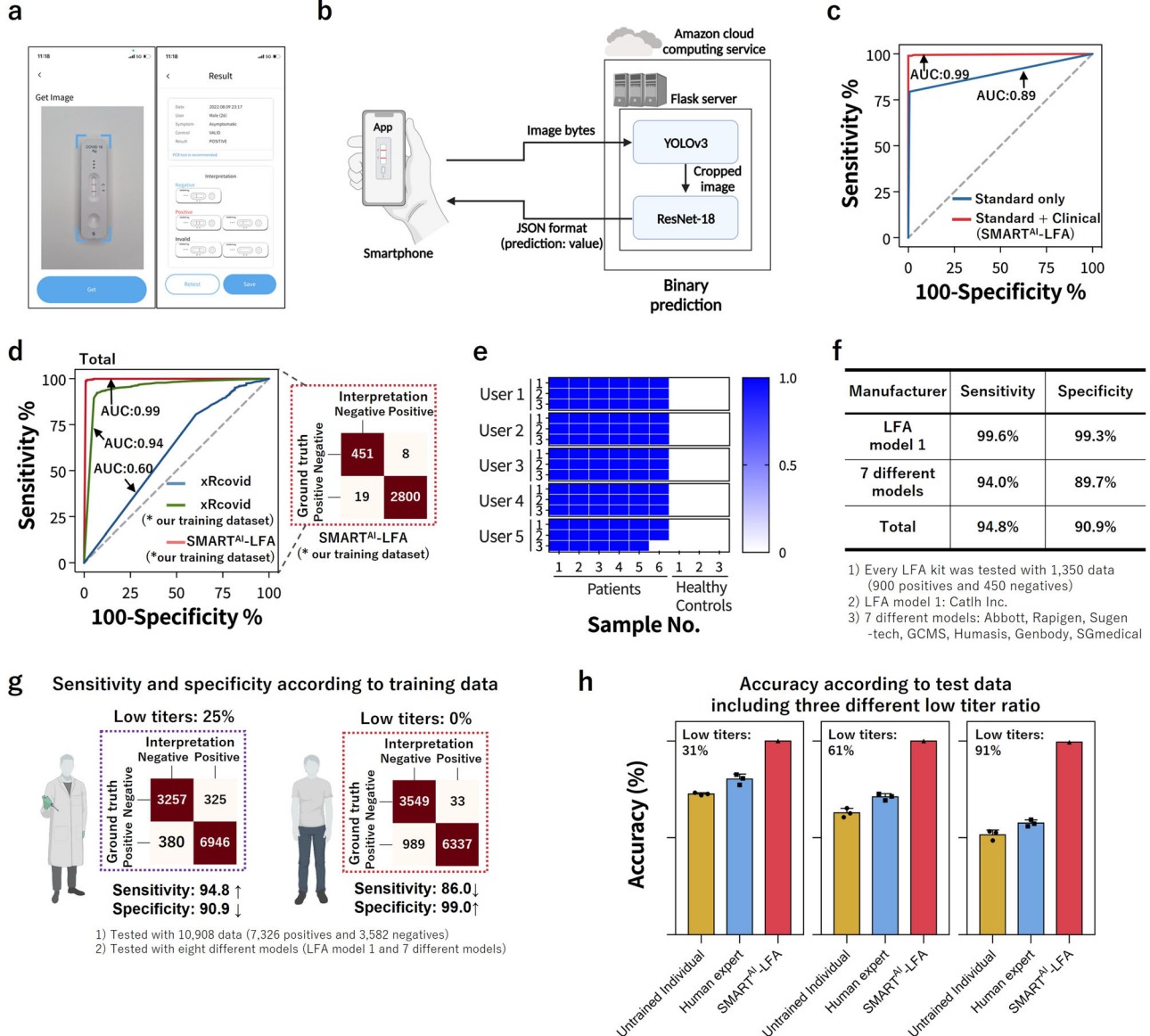

**Fig. 4 | Clinical tests with a smartphone application. a** Smartphone application and **b** schematic diagram depicting the data flows from a smartphone to the server (AWS), where the two algorithms are located. **c** The ROC curves according to training data (standard only (*n* = 8914) and additional clinical data (*n* = 8005)). **d** The ROC curves and confusion matrix of the two algorithms (SMART[AI]-LFA and xRcovid) for clinical tests (*n* = 3278). **e, f** Universality test. **e** The accuracy of the 135 app-based tests with different users/smartphones, showing 98% accuracy. **f** The total averaged sensitivity and specificity from eight different LFA models (LFA

model 1 and 7 different models) were determined as 94.8% and 90.9%, respectively. **g** The tunability of sensitivity and specificity according to the training data. **h** The accuracy according to test data (*n* = 3), including three different low-titer ratios (31, 61, and 91%), showing more reliable performance of SMART[AI]-LFA than humans. Error bars represent standard deviation from the mean. AWS Amazon Web Services, ROC receiver operating characteristic, SMART[AI]-LFA deep learning-assisted smartphone-based LFA.

image, which confirms that deep learning-assisted SMART[AI]-LFA can provide precision decisions superior to those of human experts.

**Prediction performance test**
To check cross-reactivity of COVID-19, we evaluated cross-reactivity using different respiratory viruses such as the respiratory syncytial virus (RSV A), influenza A (H1N1), influenza B, and human coronavirus-OC43 (HCoV-OC43) (*n* = 3 for each sample); however, we observed no cross-reactivity (Fig. 3f). We analyzed the concentration prediction ability of SMART[AI]-LFA using a heat map to explore the feasibility of the quantitative analysis (Fig. 3g). The concentration prediction was conducted by modifying the cost function and the output node of the classification model. Further, for the concentration, the annotated version of the dataset was used. From the SMART[AI]-LFA test with

different nucleocapsid (N) protein concentrations and its prediction, we depicted a heat map with a navy blue-red-yellow color continuum. The heat maps indicates that the prediction made by our model exactly corresponds with the real concentration of the nucleocapsid (N) protein, which implies that SMART[AI]-LFA can provide a platform not only for the binary test but also for quantitative analysis. We observed that the $R^2$ values reach 0.99 from the graph of true concentration versus predicted concentration, showing excellent sample-to-answer ability (Supplementary Fig. 3). In addition, we determined the limit of detection (LOD) for untrained individuals, human experts, and SMART[AI]-LFA as 1.25 ng/ml, 0.625 ng/ml, and 0.156 ng/ml, respectively. We set LOD values under the manufacturer's guidelines with 19/20 criteria (See more details in the method section). The LOD with SMART[AI]-LFA is enhanced up to 8-fold compared to that for untrained

individuals, which confirms its applicability to the AI-based sample-to-answer platform.

We present the prediction ability of virus titers/concentrations in clinical samples (Fig. 3h). We prepared a clinical sample from a patient (female, 33 y), and then prepared several diluted samples. The normalized concentration decreased with the dilution factors. Analyzing the concentration prediction ability implies that SMART[AI]-LFA can provide a platform for quantitative analysis.

In Fig. 3i, we showed the advantage of daily monitoring. Daily monitoring of COVID-19 could be a crucial test for early detection and convalescence monitoring. To show the feasibility of daily monitoring of COVID-19, we collected daily samples from an infected individual with symptoms (male, 27 y) (Fig. 3i). We collected samples from the individual with the symptoms and the day of onset (day 0) was defined as the first day of a symptom. The patient received the result of RT-qPCR positive on the first day of symptoms. We acquired daily data with RT-qPCR and LFA, then showed the Ct value (top) and prediction level (bottom) with days. The prediction level was based on the results of concentration prediction (Fig. 3g), classifying the prediction level (0 to 12) according to concentration (0 to 5 ng/ml). Frequent, highly sensitive LFA testing (2 to 3 daily tests) can provide the on/off signal, even the quantitative data. Daily monitoring data have revealed that the prediction level of SMART[AI]-LFA is highly correlated with the viral load (Ct value), indicating that our technique will be able to evaluate the frequency test with much higher accuracy than the commercial LFA and RT-qPCR.

## Clinical tests with a smartphone application (App)

Figure 4 shows the clinical tests conducted using a smartphone application that employs the sample-to-answer strategies. We designed a smartphone application to assist decision making when validating an AI-assisted smartphone with the developed algorithm (Fig. 4a, b). Figure 4a shows the smartphone applications (apps); we developed an AI-assisted assay based on smartphone apps using cloud-based deep learning (Supplementary Movie 1). Since the distance between the camera of smartphone and LFA test kit might influence the image size and quality, we provided a guideline (blue line in Fig. 4a) for image capture processing. Following these guidelines, individual users can capture the images without considering the distance between the camera and the LFA test kit.

Figure 4b shows a schematic diagram depicting the data flows between smartphones and servers. The image of the test results acquired from a smartphone are sent to a server (Amazon Web Service, AWS) with image bytes, and the test results determined by the deep learning algorithm are transmitted back in the JSON format to the smartphone applications. We imported the Flask module, a micro web framework written in Python, of SMART[AI]-LFA and created a Flask web server on Amazon Cloud. CNN models with YOLOv3 and ResNet-18 are used to realize the sample-to-answer platform of the COVID-19 POCT. LFAs generally require assay times of up to 15 min. In app operation, the additional time needed from taking the image to returning the result is generally within tens of seconds, depending on the network and smartphones.

In Fig. 4c, we illustrated the clinical training data effect on the ROC curves, representing accuracy. First, we depicted the ROC curves trained with standard data set ($n = 8914$, blue line, standard only), and then showed the enhanced ROC curves trained with additional clinical data ($n = 8005$, red line, standard and clinical). The ROC curve is used to evaluate the clinical ability of the diagnostic models[45]. The ROC curve indicates how well the model separates individuals into two classes; this provides information on the AUC, which measures the accuracy of the diagnostic test. The AUC trained with additional clinical data (0.99) is higher than that trained with standard dataset (0.89). SMART[AI]-LFA cannot meet the accuracy requirements of the laboratory test like RT-qPCR; however, it can continuously enhance its diagnostic accuracy by additional learning from the acquired images. Therefore, we can increase clinical accuracy with additional deep learning using clinical samples.

We represent the ROC curves and confusion matrix of the two algorithms (SMART[AI]-LFA and xRcovid) from the clinical samples (Fig. 4d). The xRcovid is a published algorithm[21] of artificial intelligence to improve COVID-19 rapid diagnostic tests and opened on Github (https://github.com/dmendels-collab/xRcovid) and Zenodo (https://zenodo.org/badge/latestdoi/312230700). To check the effect of training data and algorithm models, we prepared three cases: SMART[AI]-LFA (our training set), xRcovid (our training set), and xRcovid. To assess the diagnostic accuracy of the models, we trained two models (SMART[AI]-LFA and xRcovid) with our training set, which included the training data set from standard ($n = 8914$) and clinical data set ($n = 8005$). Further, we compared it with the xRcovid algorithm with its pre-trained model[21]. We prepared 3278 clinical tests (positive $n = 2819$ and negative $n = 459$). The AUC reveals an enhanced accuracy (0.99) compared to that of xRcovid with our training set (0.94) and xRcovid (0.60) for clinical samples. The sensitivity and specificity of SMART[AI]-LFA from the confusion matrix are 98.7% and 97.8%, respectively, which are notably higher than those of xRcovid with our training set (91.7% and 84.0%) and xRcovid (94.2% and 19.0%). The difference in AUC and accuracy between SMART[AI]-LFA and xRcovid (our training set) indicates the algorithm's ability. In addition, the larger differences in AUC and accuracy in xRcovid (our training set) and xRcovid indicate the effect of training data in clinical tests.

An important parameter of the AI-assist app is universality. We confirmed the universality by validating the ability of multi-users and multi-LFA models (Fig. 4e, f). First, five smartphone users with different smartphone models (LG Q51, Galaxy A52, iPhone 12 mini, iPhone 11 Pro, and iPhone 14 Max; Supplementary Table 4) tested the smartphone app-based diagnostics for multi-user tests under various surroundings such as indoors/outdoors, lighting conditions, and shade/sunlight with various backgrounds (Fig. 4e, Supplementary Fig. 4 and Supplementary Movie 2). Every user took three images via a smartphone app from the LFA tests of nine clinical samples (6 positives for COVID-19 and 3 healthy controls), and then acquired the results. The accuracy of the 135 app-based tests with different users/smartphones was determined as 98%.

Second, we validated the universality by testing an additional seven different LFA models (Fig. 4f, Supplementary Fig. 5, Supplementary Table 5, and Supplementary Movie 3). We used Panbio COVID-19 Ag, BIO CREDIT COVID-19 Ag, SGT-flex COVID-19 Ag, GENEDIA COVID-19, Humasis COVID-19 Ag Test, Genbody COVID-19 Ag, Insta-View COVID-19. Note that we carried out no additional training, meaning that we used the SMART[AI]-LFA algorithm trained with LFA model 1 (COVID-19 Ag LFA kits, Calth Inc.). Then, we validated SMART[AI]-LFA using three LFA models ($n = 360$, Rapigen, SD biosensor, and Yuhan, Republic of Korea, see Supplementary Table 5) and finally tested seven different models with smartphone app-based image acquisition ($n = 9450$). Every LFA kit was tested with 1350 data (900 positives and 450 negatives). To avoid overfitting during the learning process, we tried to validate the model using the different LFA models from different manufacturers used in learning and testing. The sensitivity and specificity of Model 1 (Calth Inc.) from the confusion matrix are 99.6% and 99.3%, respectively. Interestingly, the average sensitivity and specificity with seven different LFA models from 9450 app-based tests were 94.0% and 89.7%, respectively. The total averaged sensitivity and specificity from eight different LFA models (LFA model 1 and 7 different models) were determined as 94.8% and 90.9%, respectively, indicating good universality of SMART[AI]-LFA.

The aim of Fig. 4g has been to identify the tunability of sensitivity and specificity. Generally, LFA manufacturers control the sensitivity and specificity of commercial LFA by optimizing chemistry, materials, and LFA design. To determine whether the training data is associated

with the tunability of sensitivity and specificity, we controlled the ratio of the low titers in the training data. We trained the algorithm with the existing dataset (SMART$^{AI}$-LFA) and validated ($n = 360$, Rapigen, SD biosensor, and Yuhan, Republic of Korea, see Supplementary Table 5) and finally tested 10,908 data (7326 positives and 3582 negatives). We visualized the confusion matrix, which allows the extraction of the true negatives (TN), true positives (TP), false negatives (FN), and false positives (FP). The true positive and true negative portions of the confusion matrix indicate the sensitivity and specificity of the clinical assay, respectively. With more low titer (high/middle/middle-low/low titer = 20/25/30/25%) of test data, we acquired higher sensitivity for eight different LFA models (LFA model 1 and 7 different models) as 94.8% with 90.9% specificity. Meanwhile, the algorithm trained without low titer samples (high/middle/middle-low/low titer = 27/33/40/0%) yielded higher specificity (99.0%) with less sensitivity (86.0%).

Finally, we showed the accuracy according to test data, including three different low-titer ratios (31, 61, and 91%) (Fig. 4h). We designed blind tests with 150 data (100 positives and 50 negatives) for each untrained individual ($n = 3$) and human expert ($n = 3$), then compared it with SMART$^{AI}$-LFA. We evenly distributed the tests (high/middle/middle-low) with low-titer ratios. For example, for a 31% low titer design, we prepared 23% high, 23% middle, 23% middle-low, and 31% low. Thus, we investigated the significant decrease in the accuracy of humans for lower concentrations, representing the human bias under lower concentrations. With an increase in the lower titer ratio (31 to 91%), we observed a significant decrease in the accuracy of untrained individuals (72.6 to 51.6%) and human experts (80.2 to 57.6%). Meanwhile, we noticed that the accuracy of SMART$^{AI}$-LFA is maintained at over 99.0% (100 to 99.3%). The accuracy data in Fig. 4h explicitly confirms the more reliable performance of SMART$^{AI}$-LFA in comparison to humans (untrained individuals and human experts).

The SMART$^{AI}$-LFA study had a major limitation: image quality depends on the smartphone. the high-end smartphone has an automatic filter function, which decreases data accuracy because the smartphone automatically corrects its image quality. Therefore, further studies need to focus on acquiring raw data. Another limitation is that the accuracy could be reduced if the test surroundings are outside the training data. Testing the surrounding out-of-distribution training data potentially limits the accuracy of the current deep learning model. Retest signs in the image acquisition stage can be considered a solution to address these problems. Guided by the retest sign, images can be obtained within the training-data distribution.

## Discussion

SMART$^{AI}$-LFA, with a cloud-based algorithm consisting of a YOLOv3 and ResNet-18 for object finding and classification, provided 98% accuracy via smartphone application-based clinical tests. There are four practical advantages of SMART$^{AI}$-LFA.

(1) Unlike commercialized applications such as Abbott NAVICA, SMART$^{AI}$-LFA requires no further interpretation by a human expert/physician; therefore it provides higher accurate test results without any intervention.

(2) Combining clinical data learning and two-step algorithms (object findings and classification) eliminates the need for external cradles attached to the smartphone, enabling a cradle-free on-site assay with higher accuracy than the human experts.

(3) Validation of multi-users and multi-LFA models showed the universality of SMART$^{AI}$-LFA. Since smartphone users exceed 6 billion, SMART$^{AI}$-LFA with smartphone can improve the POCT along with affordability.

(4) SMART$^{AI}$-LFA can continuously increase its diagnostic accuracy by additional learning data from clinical tests and meets the new criterion for ideal REASSURED diagnostics with digitalized real-time connectivity.

A small improvement in accuracy (i.e., sensitivity and specificity) can impact not only the care and treatment of individual patients but also public health in terms of patient quarantine and disease control since a great number of COVID-19 tests are performed daily. Deep learning-assisted decision architectures allow training to be continuously enhanced, which can help us improve the decision-making capabilities over time. The availability of multi-user and multi-model systems with a reduced bias for low-titer samples provides a promising avenue for achieving breakthroughs in the early detection of infections using SMART$^{AI}$-LFA technology. By circumventing the reading ambiguities encountered in LFA tests, we anticipate that SMART$^{AI}$-LFA will be particularly effective in detecting infections in their early stages.

## Methods

### Ethical Statement

Samples were prospectively collected from patients diagnosed with COVID-19 infection at Seoul St. Mary's Hospital from April 2021 until May 2022. This study was approved by the institutional review board (KC21TIDI0134K) at Seoul St. Mary's Hospital, and informed consent was obtained from the participants.

### Commercial LFA and assay (positive, negative, and invalid)

Commercial COVID-19 Ag LFA kits (Calth Inc., Republic of Korea) are used. Following the guidelines of the manufacturer, we mixed the sample (10 µl) with running buffers (90 µl), dropped 3–4 drops onto the sample reservoir, waited for 15 min, and then acquired digital images using smartphones (Supplementary Fig. 6). We used three different smartphones (two androids: Samsung Galaxy S21 and M21, and one iOS: Apple iPhone SE) to acquire better training performance.

For the standard samples, COVID-19 nucleocapsid protein recombinant antigen (45 kDa, FPZ0516, Fapon Biotech Inc., China), which is known to be the best COVID-19 Ag test target, is prepared using 1×PBS buffer (LB004, DUKSAN, Republic of Korea). Seven different concentrations of COVID-19 nucleocapsid protein samples are prepared with 1×PBS.

The samples of COVID-19 patients and healthy controls collected from Seoul St. Mary's Hospital with the appropriate Institutional Review Board Committee approval (KC21TIDI0134K) were used as the datasets for the clinical tests. nasopharyngeal/oropharyngeal (NP/OP) swabs and saliva samples from COVID-19 patients were separately prepared in a viral transport medium (VTM) and a sterilized tube. The samples were then diluted with PBS. All NP/OP and saliva samples are labeled using RT-qPCR. For RT-qPCR, we used an In-house protocol developed by the Institute Pasteur (Paris), one of the WHO reference laboratories. Based on this protocol, we used SuperScript™ III Platinum® One-Step Quantitative RT-PCR System (Invitrogen) and Light-Cycler 480 real-time PCR machine (Roche).

We serially diluted NP/OP samples with PBS and then prepared several concentrations of clinical samples (up to 0.1×-folds dilutions) to test their ability to sense a low viral load. Like NP/OP, saliva samples with a high viral load are serially diluted with PBS buffer, and then, several concentrations containing low viral samples (Ct > 30) are tested. For healthy controls, we collected (or purchased) samples from healthy volunteers and stored them in −20 °C freezers. We collected respiratory syncytial virus (RSV A), influenza A(H1N1), influenza B, and Human coronavirus-OC43 (HCoV-OC43) samples from Seoul St. Mary's Hospital to evaluate cross-reactivity.

For invalid test results, we deliberately fabricated an LFA kit without a control line by collaborating with the manufacturer of the LFA (Calth Inc., Republic of Korea) (Supplementary Fig. 7). We acquired training data ($n = 3076$) for invalid categories, and trained all cases by adding invalid cases as third cases, which included positive, negative, and invalid cases.

We tested the accuracy of the blind test using data sets ($n = 1500$) from clinical samples. To calculate the LOD values, we prepared a

second dataset ($n = 34$: 26 positives and 8 negatives) with different concentrations, then set the LOD values under the manufacturer's guidelines with 19/20 criteria, which represents the LOD is determined as the lowest concentration where 95% (19/20) are positive (https://www.fda.gov/media/137302/download). We evaluated the accuracy of the blind test using a dataset of clinical samples ($n = 1500$). To determine the limit of detection (LOD), we prepared a second dataset ($n = 34$: 26 positives and 8 negatives) with varying concentrations. We then determined the LOD values using the manufacturer's guidelines, which indicate that the LOD is the lowest concentration at which 95% (19/20) of the tests are positive (as outlined in https://www.fda.gov/media/137302/download).

### Smartphone and data acquisition

Digital images were acquired using the three smartphones. We acquired all images under controlled light conditions by controlling the light brightness (0, 30, 60, and 90%) and light temperature (3000, 4000, and 5000 K) for the first dataset of standard data (Supplementary Fig. 8). We trained 8914 (positive: 5801 and negative: 3113) for the first dataset (1st data set: standard data). For the second dataset, we used clinical samples and prepared 8005 COVID-19 training data (5667 positives and 2338 negatives) and 1458 test data (1026 positives and 432 negatives). For the multi-model test, we prepared 9450 test data (900 positives and 450 negatives for each model) and assayed Panbio COVID-19 Ag (Abbott, USA), BIO CREDIT COVID-19 Ag (Rapigen, Republic of Korea), SGT-flex COVID-19 Ag (Sugentech, Republic of Korea), GENEDIA COVID-19 (GCMS, Republic of Korea), COVID-19 Ag Test (Humasis, Republic of Korea), COVID-19 Ag (Genbody, Republic of Korea), InstaView COVID-19 (SGmedical, Republic of Korea).

### Data for deep learning models

We prepared datasets for training, validation, and testing. The training dataset was used to train the model. The validation dataset was used to determine when to stop training by validating the model in the middle of training; here, "valid" implies that the model is not overfitted. The test dataset is used to evaluate the performance of the trained model. The test dataset consists of data that are not used for training and they include many data collected in other environments for testing. A random 10% sample of the test dataset is used as the validation dataset; the training termination condition is set to the point at which the accuracy of the validation dataset is the highest. The validation dataset is used only for validation and not for training. For object finding, 1295 training data and 100 test data are prepared. Unlike the original YOLOv3 task, class information is excluded from the label because the task in this stage is to crop the test line area; the class score term in the loss is also excluded. Two data types are constructed for the decision stage for the identified positive/negative classification: positive/negative classification data and concentration prediction data. The binary classification data comprises 16,919 training data and 1458 test data. The concentration prediction data comprises 12,964 training data and 3149 test data.

Next, we trained under various surrounding conditions where the images are captured (indoors/outdoors, different lighting conditions, and shade/sunlight). First, we prepared the data from three smartphones with a single lighting condition followed by data augmentation (augmented SL), multiple lighting conditions, and data augmentation (augmented ML). Finally, we added indoor/outdoor light conditions, day/night, various backgrounds, and shadows, and then performed data augmentation (all surroundings). For the clinical evaluations, we retrained the algorithms using real clinical samples ($n = 8005$) from COVID-19 patients ($n = 45$) and healthy controls ($n = 20$) (i.e., SMART$^{AI}$-LFA). For the training, we diluted the clinical samples with 1×PBS, and then prepared samples with different virus titers. We tested the clinical samples of NP/OP and saliva from COVID-19 patients.

### Deep learning models

The structure of the model includes two stages: the region of interest (ROI) crop stage and the decision stage. The model structure is divided into two stages because meaningful information in the original image is concentrated in a very small area near the test line, as shown in Fig. 2a. It is inefficient to make a decision using the original image directly because it has a small amount of information compared with the image size. Here, we focus on the fact that humans only look closely at the test line area when they use a diagnostic kit. Learning efficiency can be increased by imitating this human behavior pattern, i.e., by separating the task that accurately crops the test line area (ROI crop stage) from the task that performs diagnosis using the information of the test line (decision stage).

A modified version of YOLOv3[46], which is a well-known object finding model, is employed for the ROI crop stage. The sum of three loss functions is used as the loss function for this stage: confidence, localization, and classification losses. Confidence loss is calculated by comparing the objectness score value in the output vector with the existence of the actual object. The localization loss represents the extent to which the x, y, width, and height of the output vector differ from the actual value; the larger the difference, the larger is the loss. The classification loss is a value that indicates the accuracy of the predicted object class. Loss is calculated by comparing the class score of the output vector with the actual class value.

For the decision stage, a model with multiple convolutional layers is designed for the LFA kit diagnostic task. Models with similar structures were prepared for two types of outputs: a binary (positive/negative) or real number (concentration prediction); each model was trained separately using different training data. The ResNet-18 network[42] was used for positive/negative classification, and the ResNet-50 network[42] was used for concentration prediction. The binary cross entropy (BCE, Eq. 1) and root mean square error (RMSE, Eq. 2) were employed as loss functions for the binary classification and concentration prediction.

$$\text{BCE} = -\frac{1}{N}\left(\sum_{i=1}^{N} y_i \log(\text{h}(x_i\theta)) + (1 - y_i)\log(1 - \text{h}(x_i\theta))\right) \quad (1)$$

$$\text{RMSE} = \sqrt{\frac{1}{N}\sum_{i=1}^{N}(y_i - \hat{y}_i)^2} \quad (2)$$

Note that $N$ = number of training datasets, $y_i$ = binary classification truth value of $i \to 0$ or 1, $x_i$ = input data of $i$, $\theta$ = parameter of model, h = model, $\text{h}(x_i\theta)$ = probability prediction of input $x_i \to [0,1]$ with 0 to 1 value, $y_i$ = density truth value, $\hat{y}_i$ = density prediction.

### Data augmentation and analysis

A physical sample is photographed several times while modifying the lighting and shooting locations. Then, one image is regenerated into multiple images by randomly combining the following techniques: (A) color temperature modulation, (B) contrast modulation, (C) brightness modulation, (D) Gaussian blur, (E) horizontal reverse, and (F) stochastic cropping and resizing. These techniques are employed considering possible situations in which an actual user takes a photo of the LFA kit using a smartphone. A, B, and C indicate that the color implementation of each smartphone differed; C is used to consider various light conditions around the LFA kit; D is required to arbitrarily create an out-of-focus situation in the photo; and E and F reflect minute errors in the object finding process.

A confusion matrix visualizes and summarizes the performance of a classification algorithm, which allows the extraction of the probability of true negatives (TN), true positives (TP), false negatives (FN), and false positives (FP). Further, we use data augmentation to improve

the convergence of the deep learning models by varying the light luminance (from 50% to 200%) and temperature (from 4000 K to 16,000 K). Error bars in the figures represent mean ± SD. Data are analyzed using Microsoft Excel, Prism v 7.0 and BioRender software for graphical analysis.

### Smartphone application via cloud-based deep learning

We developed a smartphone-based application for both Android and iOS systems. We designed apps for end users to send a picture of their COVID-19 Ag test results and obtained the results. To use AWS, we imported the program onto Amazon EC2 and used Python with Flask, which is a web framework written in Python.

We implemented the deep learning model using Ubuntu 16.04 as the OS, Python as the programming language, and PyTorch as the deep learning library. The batch size was set to 64, subdivision 16, width 416, height 416, channels 3, momentum 0.9, decay 0.0005, learning rate 0.001, max batches 4000, steps [3200, 3600], and class 1 when training the object findings.

The batch size and subdivision are parameters that determine the number of images to push on the GPU at once (mini-batch size = batch/subdivision). The width and height indicate the size required to initially resize the input image. The channels represent the number of channels in an image. The momentum represents the weight of how much inertia is maintained in the optimization process, and the decay and steps are parameters that decrease the learning rate by decay and learn at the corresponding step (epoch). Finally, classes represent the number of data classes to be trained.

Most parameters are set to be the same as object findings when training the decision stage. The learning rate and max-batches for the positive/negative classification tasks are modified to 0.0001 and 1000, respectively. The concentration prediction task batch size, learning rate, and maximum number of batches are modified to 16, 0.0001, and 200, respectively.

### Reporting summary

Further information on research design is available in the Nature Portfolio Reporting Summary linked to this article.

## Data availability

All data supporting the findings described in this manuscript are available in the article and in the Supplementary Information and from the corresponding author upon request. Example images used in this study are available at (https://drive.google.com/file/d/16Rv9rcavScqK7UFFjZFcgys4vE3aZzn8/view?usp=sharing) Source data are provided with this paper.

## Code availability

The overall source codes used in this study is available at: (https://github.com/Artinto/Sample-to-answer_COVID-19). Android's smartphone application (App) can be downloaded at: (https://drive.google.com/file/d/1zyp5I5q8dpqshWo1HhaTG8iBwfmqC6MI/view?usp=sharing).

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

## Acknowledgements

This research was supported by the Bio & Medical Technology Development Program of the National Research Foundation (NRF) funded by the Korean government (MSIT) (No. 2021M3E5E3080743). The work reported in this paper was conducted during the sabbatical year of Kwangwoon University in 2020 (K.B.L.). Illustration in Figs. 1, 2a, 3a, 4b, d and Supplementary Fig. 1 were created with BioRender.com.

## Author contributions

S.L., S.K., Y.K.Y., K.B.L., and J.H.L. conceived and designed the study. J.S.P. and H.W. acquired the images. S.K. and H.W. were responsible for the programming. S.C. and C.P. provided the clinical samples and interpreted the results. D.L. provided the kit. S.L., S.K., D.S.Y., Y.K.Y., K.B.L., and J.H.L. wrote the manuscript. All the authors discussed the results and commented on the manuscript.

## Competing interests

The authors declare no competing interests.
