## [Peer Review File · Nature Communications]

Reviewers' comments:

Reviewer #1 (Remarks to the Author):

Submission ID.

NCOMMS-22-39189-T

Title

Sample-to-answer Platform for the Clinical Evaluation of COVID-19 using a Deep-learning-assisted Smartphone-based Assay

Authors

Lee, Seungmin; Department of Electrical Engineering, Kwangwoon University; School of Biomedical Engineering, Korea University, Seoul, Republic of Korea

Kim, Sunmok; Department of Electrical Engineering, Kwangwoon University, Seoul, Republic of Korea

Park, Jeong Soo; Department of Electrical Engineering, Kwangwoon University, Seoul, Republic of Korea

Woo, Hyowon; Department of Electrical Engineering, Kwangwoon University, Seoul, Republic of Korea

Lee, Dongho; CALTH Inc., Gyeonggi, Republic of Korea

Yoon, Dae Sung; School of Biomedical Engineering, Korea University, Seoul, Republic of Korea

Cho, Sung-Yeon; Vaccine Bio Research Institute; Division of Infectious Diseases, Department of Internal Medicine, College of Medicine, The Catholic University of Korea, Seoul, Republic of Korea

Park, Chulmin; Vaccine Bio Research Institute, College of Medicine, The Catholic University of Korea, Seoul, Republic of Korea

Yoo, Yong Kyoung; Department of Electronic Engineering, Catholic Kwandong University, Gangwon-do, Republic of Korea

Lee, Ki- Baek; Department of Electrical Engineering, Kwangwoon University, Seoul, Republic of Korea

Lee, Jeong Hoon; Department of Electrical Engineering, Kwangwoon University, Seoul, Republic of Korea

Summary

Lee, Jeong Hoon and co-authors describe the development of deep-learning assisted smartphone-based LFA system for testing severe acute respiratory syndrome coronavirus 2 (SARS-CoV-2) virus. Authors report a significantly enhanced performance over both trained and untrained users and a commercially available algorithm that was developed for the same purpose (i.e., xRcovid). The use of clinical samples and testing and monitoring actively infected patients are among the strengths of this study. The

developed system is simple, easy to use, and might be useful for frequent, self-testing of COVID-19 and other respiratory infections.

General editorial comments

This manuscript is well-written and presented. The topic is interesting and appropriate for the journal, balancing overview with details. The detailed descriptions of methods and results of the developed system are instructive and accurate. However, the discussions and the selection of references provides a biased overview, and there a significant body of published articles in this direction is not included or fairly presented.

The figures, in addition, do not yet match the expected quality for publishing in Nature Communications. Figures need to be improved, and simplified, with adding text to ensure the overall highly instructive character of this article. Each term used in figures or figure captions must be clearly defined. There are some additional issues and questions (itemized below), which I encourage the authors to address to improve the quality of this manuscript before publication.

Strengths

- Clear and concise content
- Well designed and performed study
- The use of clinical data set
- Extensive and instructive descriptions throughout the manuscript.

General weaknesses

- The originality of the work is compromised by previously published articles.
- The development of a hardware-free system versus a hardware-dependent system for LFA is not necessarily significant. Phone attachment can be more useful than having an algorithm that is limited to the used type of LFA device, probably due to the use of object finding/recognition.
- The clinical use of this system is unclear and not well justified. Although, the reported accuracy is > 99%, what are the diagnostic recommendations after using this system? Is it significantly different than using other LFA systems? For instance, what is the recommended resolution when there is a conflict between a clinician and the AI-LFA system?
- The study is missing a detailed comparison with a 'gold standard' technique currently used in diagnosing COVID-19, i.e., RT-qPCR or other NAT systems.

Specific weaknesses

- Needs a major effort for editing towards improving the writing flow, simplified terminologies, and a smaller number of abbreviations.
- How your system performed at low virus titers? It can be useful to present these data in comparison to other tested algorithms and assays.
- Although the algorithm combining the two models sounds new, it presents a limitation to the developed system. The training and testing sets used for this study are focused on LFA devices produced by a single provider, without considering the variations that already exist in LFA systems' material, structure and dimensions. Other hardware-free systems are there, reporting a higher detection efficiency and sensitivity.
- Need to define what is a standard sample at the first time mentioned in the manuscript.
- What was the 'gold standard' method used to perform the blind test, was the PCR or target spiked samples? Would this test eliminate the need for PCR? If a test result was not approved by an expert human, what would be its significance from the healthcare point of view?
- The advantage of developing a cradle-free smartphone system is debatable for LFA. LFA device might just need optic-free and inexpensive attachment for accurate imaging. I would revise the statement in Line 83.
- P5 L98: Untrained individuals use the NOVICA app to obtain the test results; However, the decision is made by human experts, not by AI. Why do authors consider this as a drawback or limitation? Would AI be solely used for deciding and recommending treatment?
- P5 L104, the blind test was performed with $n = 82$, but only 62 samples are reported in the paragraph. Please check and correct it to reflect the right number.
- The blind test or other clinical tests were performed for pre-diagnosed or post-diagnosed individuals/samples, with and without symptoms. Any more information about the virus titer?
- Classification or classifications?
- P4 L78, 'Results' should be 'Results and Discussions.'
- P5 L115, define what are 'real measuring conditions.' This statement is unclear: several undefined terms are used e.g., real diagnostics, severe and critical cases, and the test conditions.
- Throughout the manuscript, authors shift from using the term sample to the term data. It is highly recommended that authors mention the actual number of samples and the extracted data points used for training, testing and validation sets.
- P6 L125, 126, what are the standard training data, and the mentioned number of 8,914 represents samples or data points, and how many samples are used?
- Untrained individuals (59.5–100%) and human experts (79.5–99.3%), this difference in the results of ROC analysis can indicate bias in the data set used for this comparative testing. Detailed data are needed.

- P9 L199, was the LOD estimated using the same set of LFA devices/samples?
- The manuscript might benefit from a schematic diagram that presents the different steps and stages followed for app development.
- What was the used RT-PCR kit, and any specific reason for using this specific kit?
- An invalid test can be because of the absence of the control line, but also due to the failure of reagents, and defective device, was this addressed during the development of the algorithm?
- It would be useful to add a table that presents the data points or samples, and how each group (algorithm vs trained vs untrained) this will give a chance to for evaluating where the algorithm performed better. An image of the detection zone of the LFA device can also be very informative to analyze this performance.
- The full name of AUC and its abbreviation should be fixed throughout the manuscript.
- It is unclear why the authors used DL v2 in clinical tests, and how the comparison with a more trained version of DLv3 is significant. A comparison should be performed versus the 'gold standard' technique, probably qPCR, as it is the currently used technique for diagnosing infection with SARS-CoV-2.
- How do the authors explain that the SMARTAI-LFA app is over-performing xRcovid? The explanation provided is biased, considering that the app was developed with a single and specific LFA device, while xRcovid was developed using another LFA device. Again, providing a detailed table that presents data points is necessary for studies that focus on algorithm development. It will allow the audience to evaluate the developed algorithm and the best use of it.
- What is the 'predicted level' in Figure4 f, is it based on the sample line intensity? The numbering and the use of commas and hyphens in Figure captions need to be revised.

Reviewer #2 (Remarks to the Author):

Comments: Major revision

In this paper, the authors reported a deep-learning-assisted smartphone-based LFA (SMARTAI-LFA) strategy for evaluation of COVID-19 via combining clinical data learning and a two-step CNN model (object findings and classification). With the new coronavirus raging around the world, how to use cheap test strips to get better accuracy does have great significance. However, from a technology development perspective, combining lateral flow analysis with AI, there are still many issues to be discussed. In addition, with all the validation of the technique and optimisation of the assay based on the technique, there are still a few comments that I would like to point out. In summary, I would recommend the article being considered for publication after the below concerns are addressed.

1. From a technology development perspective, with the aid of AI, the low-cost, user-friendly characteristics of lateral flow assays are weakened. After all, some poor areas/conditions do not have a smartphone/internet and can only detect the results by human eye observation. In other words, what is the significance brought about by the addition of AI to such a disposable rough testing technique? What are the implications of this for the development of future rapid tests? Please authors supplement the relevant viewpoint in introduction.

2. Is visual observation bias the biggest cause of false-positive and false negative problems in lateral flow assays? In this work, false negatives due to visual bias are greatly reduced by AI image processing, so how is the problem of false positives resolved by AI? Please give the relevant discussion in paper.

3. In result section, the authors stated that “We demonstrated the accuracy of the blind test (n = 82) using untrained individuals, human experts, and SMARTAI-LFA. Clinical samples (n=62, COVID-19 patients:42 and healthy controls: 20) that included nasopharyngeal/oropharyngeal (NP/OP) swabs and saliva samples” I think the authors should provide a table to show the SMARTAI-LFA’s individual accuracy of the blind test of Clinical samples (n=62, COVID-19 patients 42 and healthy 20; and healthy controls: 20).

4. Commercial COVID-19 Ag LFA kits from different supplier have different the degree of coloring after loading COVID-19 sample solution. In order to verify the general applicability of the developed method, I suggest the authors choose LFA kits from more suppliers for deep-learning process.

5. In order to show the method’s reliability, as known, different people using different smart-phone and taking pictures in different environments. I suggest the authors should collect more than 100 different person’s pictures using their own smartphone then using SMARTAI-LFA to make corresponding analysis, and compared with their clinical COVID-19 results, and provide the testing accuracy that will be very meaningful.

6. What’s the connection of YOLOv3,RESNET-18 and dl.v3 should be explain clearly. What improvements between versions should be further elaborated.

7. About cloud-based AI (AWS), I can’t find where to import COVID-19 photos. Is it free or paid? If the authors can provide a webpage entrance of SMARTAI-LFA to let people make a test that would be more convincing and meaningful.

8. As for the data augmentation for improving the performance, it might also be worth to consideration of the photoing distance. Because the current smart phones are autofocus, different photoing distances maybe lead to colors not be completely consistent.

9. If the proposed SMARTAI-LFA strategy is used for clinical diagnosis to COVID-19 in POCT. How is the reproducibility for different user?

10. The time from dropping sample to show assay results in smartphone should be provided.

11. In Eq. 1 and Eq. 2, the meaning of each parameter needs to be explained specified.

12. The one most important factors of biological assays is limit of detection (LOD). This article chose to calculate LOD based on the concentrations of nine out of ten untrained individuals/human experts correctly answered. What is the basis for defining LOD in this way?

13. There are several minor grammatical errors spotted throughout the reading, hence it is better to check carefully before next submission.

Reviewer #3 (Remarks to the Author):

General comments:

The authors identify an important opportunity for improvement of the performance of visually read lateral flow assays by using a deep learning algorithm to improve over human reader performance.

For the most part, the approaches are sound. However, the stated performance increases (97% accuracy vs 62% for untrained users) are higher than what I would expect to be possible with this method except for very low-titer samples. Based on the described methodology I am not clear to what extent the authors relied on diluted clinical vs actual low-titer clinical samples to train and validate their models. Specific comments are below on how the paper could be improved by better explaining the methodology.

Specific comments:

Line 27: "Current self-testing for COVID-19 detection suffers from low accuracy and reading ambiguities attributable to highly subjective readings performed by untrained individuals." I don't think this is a primary issue. Real sensitivity of antigen LFAs lags PCR of course, but that is primarily a function of the assay chemistry, not of the reading methodology.

Line 47: "Although the real-time polymerase chain reaction (RT-PCR) test is highly sensitive, one-time monitoring can detect viral shedding up to 17 days after the infection period": what does one-time monitoring refer to? With PCR? And is that small amount of shedding relevant for transmission?

Line 76 - provide reference for REASSURED.

Line 92. Describe "augmented dataset". Explain why it does not introduce bias and if and why it improves the accuracy of the deep learning method.

Line 94; "After qualifying the clinical samples, we trained the model with additional clinical samples (n = 8,005) and acquired 99.2% accuracy when using the smartphone application." How does this compare to visually read accuracy with the same data set? Also, the NAVICA app (note spelling!) is not an AI reader, it just captures images. I do not understand the relevance of the sentence starting in line 96.

Line 100 - which commercial LFA?

Line 108 - "untrained individuals, human experts, and SMARTAI-LFA to be 62.7, 76.6, and 97.9%" This enormous difference between trained and untrained users, and between trained users and the AI seems extremely hard to believe UNLESS the patient samples were strongly biased towards very low viral loads. Most patients test during the acute phase of the infection and a positive antigen line is usually clear as day. The main benefit of AI-supported reading would likely be for very weak lines, corresponding to low viral loads prior to the onset of symptoms or as the infection is waning, but that is not when most untrained users are performing home tests.

Line 158 please see my earlier comment on data augmentation

Line 213: Dilution with PBS to create low-titer samples is problematic as agents that typically cause interference and non-specific binding, especially at levels mimicking a weak positive line would also be diluted out. So results with diluted samples, for a given viral titer, will look more accurate than they would be in real undiluted clinical samples.

Line 214 - were the 83% with intermediate and low viral loads undiluted clinical samples or diluted clinical samples?

line 462 - data augmentation - it would appear that there are still possible biases from the selection of the types of variations in ambient light and imaging conditions that were selected that would create apparent improvements in accuracy that could not then be generalized widely for real use.

Reviewer #1

Summary

Lee, Jeong Hoon and co-authors describe the development of deep learning-assisted smartphone-based LFA system for testing severe acute respiratory syndrome coronavirus 2 (SARS-CoV-2) virus. Authors report a significantly enhanced performance over both trained and untrained users and a commercially available algorithm that was developed for the same purpose (i.e., xRcovid). The use of clinical samples and testing and monitoring actively infected patients are among the strengths of this study. The developed system is simple, easy to use, and might be useful for frequent, self-testing of COVID-19 and other respiratory infections.

General editorial comments

Comment 1: This manuscript is well-written and presented. The topic is interesting and appropriate for the journal, balancing overview with details. The detailed descriptions of methods and results of the developed system are instructive and accurate. However, the discussions and the selection of references provides a biased overview, and there a significant body of published articles in this direction is not included or fairly presented.

Answer) We really appreciate the reviewer's helpful comment. After carefully reading the reviewer's comments, we intensively revised all the figures, introduction, discussion, and conclusion parts with additional references.

First, we revised all the figures with additional experimental data. We revised fig.1 with simplified schemes and texts to clearly show the workflow of SMART^{AI}-LFA for COVID-19 diagnostics. We deleted the wordy text to deliver the main idea. In fig. 2, We redesigned/acquired the blind test with unbiased sample distribution (discussed later); then, we moved the data of (g) the blind test, (h) Cross-reactivity, and (i) the concentration prediction to revised fig.3. In revised fig.2, we focused only the algorithm optimization. In revised fig.3, we tried to show clinical validation via blind test, cross-reactivity, concentration prediction, and daily monitoring. We deleted the blind test of NP/OP and saliva (fig. 3(b-e) and fig. 3(f-i) in the original manuscript). Instead, we showed the blind test with increased test data from n=82 to n=1500 and an evenly distributed data set for acquiring unbiased results. Moreover, we revised fig.4 with newly added results. First, we showed the schemes of the smartphone app

and data flows from a smartphone to the server (AWS) with new flow charts (See fig.4a-b). After showing the ROC curves of additional clinical samples onto standard samples (See fig.4c), we compared two algorithms (SMART^{AI}-LFA and xRcovid) (See fig.4d). We newly added the universality by validating the ability of multi-users and multi-LFA models (fig.4e-f). Then, we show the effects of training data on sensitivity and specificity (fig.4g). Finally, we demonstrated the accuracy according to test data, showing a more reliable (unbiased) performance of SMART^{AI}-LFA than humans (fig.4h).

Second, we mainly revised the introduction parts as *“Although the real-time polymerase chain reaction (RT-qPCR) test is highly sensitive, frequent on-site tests of COVID-19 using RT-qPCR are still challenging. Generally, RT-qPCR can detect viral shedding up to 17 days after the infection period, increasing unnecessary quarantine^{2,3}. Moreover, long turnaround times of RT-qPCR allow infected people to spread the viruses exponentially before getting results.”* (See lines 47-51). and we added articles with smartphone AI as *“Since the number of current smartphone users exceeds 6 billion, meaning >80% of the world's population owns smartphones (<https://www.statista.com/statistics/330695/number-of-smartphone-users-worldwide/>), the assay using a smartphone can give accessibility and affordability. Smartphone-based assays have been performed to detect sperm concentration and motility²⁷, protein biomarkers²⁸, CRISPR-read SARS-CoV-2²⁵, Zika virus²⁹, norovirus³⁰, cell migration³¹, SARS-CoV-2 variants at single-nucleotide resolution³². To further improve the performance of smartphone assays, AI-assisted assays were performed for DNA diagnosis in malaria detection²⁴, HIV rapid tests³³, CRISPR-Cas13a based SARS-CoV-2 detection²⁶, and serological SARS-CoV-2 antibody test²¹.”* (See lines 64-73).

Third, we fully redesigned the blind test starting from the unbiased data. For this, we increased the test set number from 80 to 1,500, then evenly distributed the data set to minimize the bias of test results. We trained 8,005 images (positive: 5,667 and negative 2,338) and tested 1,500 images (positive: 1,000 and negative 500). All the blind test images are prepared from the COVID-19 patient/healthy control samples. We collected all the clinical samples from Seoul St. Mary's Hospital with Institutional Review Board Committee approval (KC21TIDI0134K) and acquired all the Ct values from the patient samples. The Ct value is not exactly inversely proportional to the color intensity of LFA test results; therefore, we prepared the blind tests set using the color chart under the LFA manufacturer's guidelines. Generally, the color chart

consisted of 10 classifications; then, we evenly prepared the blind test images (high 250 images, mid 250 images, low 250 images, and very low 250 images) to prevent bias. We revised as follows: “We determined the accuracy of the blind test using 1,500 test images from untrained individuals (n=10), human experts (n=10), and SMART^{AI}-LFA (**Fig. 1b**). Clinical samples (n=65, COVID-19 patients: 45 and healthy controls: 20) were tested to validate the clinical predictability of SMART^{AI}-LFA. We arranged the training, test and validation datasets for algorithm development (**Supplementary Table 1-2, Supplementary Fig. 1**).” (line 117-121) and “We assess the blind test using three different groups: untrained individuals, human experts, and SMART^{AI}-LFA using 1,500 test images (1,000 positives and 500 negatives). We collected clinical samples of NP/OP samples from COVID-19 patients at Seoul St. Mary’s Hospital. The information pertained to SARS-CoV-2 patients (n=45) and healthy controls (n=20) including sample collection, variants, sex, age, and Ct value (Supplementary Table. 3). All the samples were analyzed with RT-qPCR, then conducted the LFA assay. Finally, the positive data from LFA assay were classified into four groups, i.e., high/middle/middle-low/low titer, with the aid of the color chart level (high with levels 10–8, middle with level 7–5, middle-low with level 4–3, and low titer with level 2–1 for positive, and negative with level 0). For positive data (n=1,000), we evenly distributed data across four groups (n=250). We prepared negative data (health controls, n=500). For the blind test, ten untrained individuals and ten human experts tested each of the 150 test images, which included 25 high, 25 middle, 25 middle-low, 25 low, and 50 negative data. The sum of the blind tests for both untrained individuals and human experts was 1500 images.

The ROC curve shows a general overview of the three different models; a larger value of the area under the curve (AUC) indicates a better classifier. From the ROC curve in Fig. 3b, we observe larger AUCs for SMART^{AI}-LFA (1.00) compared with that for untrained individuals (0.79) and human experts (0.86); this demonstrates that SMART^{AI}-LFA is an excellent classifier for clinical assays. Fig. 3c shows the table for the three different groups, which indicates a considerable enhancement in sensitivity and specificity using a SMART^{AI}-LFA (100 and 100%) compared with untrained individuals (72.9 and 86.0%) and human experts (83 and 88.2%). To closely study the reason for the accuracy reaching 100%, we explored the effect of the training and test datasets discussed later (Fig. 4g-h).

We present three positive clinical sample images in Fig. 3d to clarify the AI’s decision ability. Fig. 3e shows the evidence of a positive test line of Fig. 3d with contrast enhancement. Although all groups can provide the correct answer as seen from the first image, only

SMART^{AI}-LFA can predict the positive samples from the third image, which confirms that deep learning-assisted SMART^{AI}-LFA can provide precision decisions superior to those of human experts.” (line 187-215)

Fourth, we relocated clinical validation of cross-reactivity, concentration prediction, and daily monitoring in fig. 3. (See revised fig.3 and manuscript parts).

Fifth, we fully revised fig. 4. We added *Supplementary Videos* to show the SMART^{AI}-LFA's operation. To depict the data flows between smartphones and servers (Fig. 4b), we revised as “**Fig. 4b** shows a schematic diagram depicting the data flows between smartphones and servers. The image of the test results acquired from a smartphone are sent to a server (Amazon Web Service, AWS) with image bytes, and the test results determined by the deep learning algorithm are transmitted back in the JSON format to the smartphone applications. We imported the Flask module, a micro web framework written in Python, of SMART^{AI}-LFA and created a Flask web server on Amazon Cloud. CNN models with YOLOv3 and ResNet-18 are used to realize the sample-to-answer platform of the COVID-19 POCT. LFAs generally require assay times of up to 15 min. In app operation, the additional time needed from taking the image to returning the result is generally within tens of seconds, depending on the network and smartphones.” (line 265-273). To check the AI-assist app's universality, we first validated the ability of multi-users with multiple smartphones (Fig. 4e-f) as “An important parameter of the AI-assist app is universality. We confirmed the universality by validating the ability of multi-users and multi-LFA models (**Fig. 4e-f**). First, five smartphone users with different smartphone models (LG Q51, Galaxy A52, iPhone 12 mini, iPhone 11 Pro, and iPhone 14 Max; **Supplementary Table. 4**) tested the smartphone app-based diagnostics for multi-user tests under various surroundings such as indoors/outdoors, lighting conditions, and shade/sunlight with various backgrounds (**Fig. 4e, Supplementary Fig. 4, and Supplementary Video 2**). Every user took three images via a smartphone app from the LFA tests of nine clinical samples (6 positives for COVID-19 and 3 healthy controls), and then acquired the results. The accuracy of the 135 app-based tests with different users/smartphones was determined as 98%.” (line 303-312). Then, we validated the universality by testing an additional seven different LFA models (Fig. 4f). “Second, we validated the universality by testing an additional seven different LFA models (**Fig. 4f, Supplementary Fig. 5, Supplementary Table. 5, and Supplementary Video 3**). We used Panbio COVID-19 Ag, BIO CREDIT COVID-19 Ag, SGT-flex COVID-19

Ag, GENEDIA COVID-19, Humasis COVID-19 Ag Test, Genbody COVID-19 Ag, InstaView COVID-19. Note that we carried out no additional training, meaning that we used the SMART^{AI}-LFA algorithm trained with LFA model 1 (COVID-19 Ag LFA kits, Calth Inc.). Then, we validated SMART^{AI}-LFA using three LFA models (n=360, Rapigen, SD biosensor, and Yuhan, Republic of Korea, See **Supplementary Table. 5**) and finally tested seven different models with smartphone app-based image acquisition (n=9,450). Every LFA kit was tested with 1,350 data (900 positives and 450 negatives). To avoid overfitting during the learning process, we tried to validate the model using the different LFA models from different manufacturers used in learning and testing. The sensitivity and specificity of Model 1 (Calth Inc.) from the confusion matrix are 99.6 % and 99.3%, respectively. Interestingly, the average sensitivity and specificity with seven different LFA models from 9,450 app-based tests were 94% and 89.7%, respectively. The total averaged sensitivity and specificity from eight different LFA models (LFA model 1 and 7 different models) were determined as 94.8% and 90.9%, respectively, indicating good universality of SMART^{AI}-LFA.” (line 313-328). In fig. 4g, we tried to determine whether the training data is associated with the tunability of sensitivity and specificity as “The aim of **Fig. 4g** has been to identify the tunability of sensitivity and specificity. Generally, LFA manufacturers control the sensitivity and specificity of commercial LFA by optimizing chemistry, materials, and LFA design. To determine whether the training data is associated with the tunability of sensitivity and specificity, we controlled the ratio of the low titers in the training data. We trained the algorithm with the existing dataset (SMART^{AI}-LFA) and validated (n=360, Rapigen, SD biosensor, and Yuhan, Republic of Korea, See **Supplementary Table. 5**) and finally tested 10,908 data (7,326 positives and 3,582 negatives). We visualized the confusion matrix, which allows the extraction of the true negatives (TN), true positives (TP), false negatives (FN), and false positives (FP). The true positive and true negative portions of the confusion matrix indicate the sensitivity and specificity of the clinical assay, respectively. With more low titer (high/middle/middle-low/low titer=20/25/30/25%) of test data, we acquired higher sensitivity for eight different LFA models (LFA model 1 and 7 different models) as 94.8% with 90.9% specificity. Meanwhile, the algorithm trained without low titer samples (high/middle/middle-low/low titer=27/33/40/0 %) yielded higher specificity (99 %) with less sensitivity (86%).”. In fig. 4h, we tried to show the effects of test data on the accuracy. “Finally, we showed the accuracy according to test data, including three different low-titer ratios (31, 61, and 91%) (**Fig. 4h**). We designed blind tests with 150 data (100 positives and 50 negatives) for each untrained individual (n=3) and human expert (n=3), then compared it with SMART^{AI}-LFA. We evenly distributed the tests (high/middle/middle-low) with

low-titer ratios. For example, for a 31% low titer design, we prepared 23% high, 23% middle, 23% middle-low23, and 31% low. Thus, we investigated the significant decrease in the accuracy of humans for lower concentrations, representing the human bias under lower concentrations. With an increase in the lower titer ratio (31 to 91%), we observed a significant decrease in the accuracy of untrained individuals (72.6 to 51.6%) and human experts (80.2 to 57.6%). Meanwhile, we noticed that the accuracy of SMART^{AI}-LFA is maintained at over 99% (100 to 99.3%). The accuracy data in Fig. 4h explicitly confirms the more reliable performance of SMART^{AI}-LFA in comparison to humans (untrained individuals and human experts).” (line 330-356).

Lastly, we added major limitation of SMART^{AI}-LFA at the end of “Results and Discussions” part as *“The SMART^{AI}-LFA study had a major limitation: image quality depends on the smartphone. the high-end smartphone has an automatic filter function, which decreases data accuracy because the smartphone automatically corrects its image quality. Therefore, further studies need to focus on acquiring raw data. Another limitation is that the accuracy could be reduced if the test surroundings are outside the training data. Testing the surrounding out-of-distribution training data potentially limits the accuracy of the current deep learning model. Retest signs in the image acquisition stage can be considered a solution to address these problems. Guided by the retest sign, images can be obtained within the training-data distribution.”* (line 357-364). Then, we added new sentences in the conclusion parts as *“(3) Validation of multi-users and multi-LFA models showed the universality of SMART^{AI}-LFA. Since smartphone users exceed 6 billion, SMART^{AI}-LFA with smartphone can improve the POCT along with affordability.”* (line 376-378)

We hope the manuscript, with careful revisions, meets your high standards. Below we provide the point-by-point responses. All modifications in the manuscript have been highlighted in red.

Comment 2: The figures, in addition, do not yet match the expected quality for publishing in Nature Communications. Figures need to be improved, and simplified, with adding text to ensure the overall highly instructive character of this article. Each term used in figures or figure captions must be clearly defined. There are some additional issues and questions (itemized

below), which I encourage the authors to address to improve the quality of this manuscript before publication.

Answer) We tried to clarify all figures with figure captions according to the reviewer's suggestion. As mentioned, we revised fig.1 with simplified schemes and texts to clearly show the workflow of SMART^{AI}-LFA for COVID-19 diagnostics. We deleted the wordy text to deliver the main idea. In fig. 2, We redesigned/acquired the blind test (n=82 to n=1500) with unbiased sample distribution (discussed later); then, we moved the data of (g) the blind test, (h) Cross-reactivity, and (i) the concentration prediction to revised fig.3. In revised fig.2, we focused only the algorithm optimization. In revised fig.3, we tried to show clinical validation via blind test, cross-reactivity, concentration prediction, and daily monitoring. We deleted the blind test of NP/OP and saliva (fig. 3(b-e) and fig. 3(f-i) in the original manuscript). Instead, we showed the blind test (increase test data from n=82 to n=1500) with unbiased sample distribution. Moreover, we fully revised fig.4. First, we showed the schemes of the smartphone app and data flows from a smartphone to the server (AWS) with its flow charts (See fig.4a-b). After showing the ROC curves of additional clinical samples onto standard samples (See fig.4c), we compared two algorithms (SMART^{AI}-LFA and xRcovid) (See fig.4d). We newly added the universality by validating the ability of multi-users and multi-LFA models (fig.4e-f). Then, we show the effects of training data on sensitivity and specificity (fig.4g). Finally, we showed the accuracy according to test data, showing a more reliable (unbiased) performance of SMART^{AI}-LFA than humans. (fig.4h).

Strengths

- Clear and concise content
- Well designed and performed study
- The use of clinical data set
- Extensive and instructive descriptions throughout the manuscript.

General weaknesses

Comment 1: The originality of the work is compromised by previously published articles.

Answer) To clearly show the originality, we first mentioned the published article of the assay using a smartphone, then showed the AI-assisted smartphone assay. Our current manuscript,

as far as our knowledge, is the first work of Point-of-care AI-based smartphone assay of SARS-CoV-2 Ag detection to face the pandemic. We tried to show our originality of SMART^{AI}-LFA by adding

“Since the number of current smartphone users exceeds 6 billion, meaning >80% of the world's population owns smartphones (<https://www.statista.com/statistics/330695/number-of-smartphone-users-worldwide/>), the assay using a smartphone can give accessibility and affordability. Smartphone-based assays have been performed to detect sperm concentration and motility²⁷, protein biomarkers²⁸, CRISPR-read SARS-CoV-2²⁵, Zika virus²⁹, norovirus³⁰, cell migration³¹, SARS-CoV-2 variants at single-nucleotide resolution³². To further improve the performance of smartphone assays, AI-assisted assays were performed for DNA diagnosis in malaria detection²⁴, HIV rapid tests³³, CRISPR-Cas13a based SARS-CoV-2 detection²⁶, and serological SARS-CoV-2 antibody test²¹.” (See lines 64-73).

For smartphone assay, two main approaches are cradle and cradle-free. Phone attachment (cradle) could be one of the great approaches for smartphone-based diagnostics. However, there are several limitations to the use of external cradles. The main issue is the universality of external cradles. Each smartphone has a different size with different camera locations, so the “one cradle fits all” strategy are not work. One needs each cradle for each smartphone. Moreover, many external cradles have their CCD for reading light signal Bluetooth for transmitting a signal to smartphones, consequently increasing the costs. We revised the manuscript with citations as *“The most significant advantage of SMART^{AI}-LFA is that it requires no external cradles attached to the smartphone. Since each smartphone has a different size and camera location, the one-cradle-fits-all approach may not work. Moreover, many external cradles need their external optics^{27, 31, 32, 34}, fluorescence components²⁵, microscope^{28, 30}, and Bluetooth^{24, 35} for reading and transmitting signals to smartphones, consequently increasing costs. Meanwhile, the use of AI could eliminate external cradles, which meets the universality of smartphone-based assays discussed later. Further, a smartphone-based AI provides a great opportunity to meet the REASSURED criteria (Real-time connectivity, Ease of specimen collection, Affordable, Sensitive, Specific, User friendly, Rapid and robust, Equipment free and Deliverable to end-users)²², which are the new criteria for digital connectivity.”* (line 88-98).

To strengthen the universality of our SMART^{AI}-LFA, we intensively revised the manuscript as “An important parameter of the AI-assist app is universality. We confirmed the universality by validating the ability of multi-users and multi-LFA models (**Fig. 4e-f**). First, five smartphone users with different smartphone models (LG Q51, Galaxy A52, iPhone 12 mini, iPhone 11 Pro, and iPhone 14 Max; **Supplementary Table. 4**) tested the smartphone app-based diagnostics for multi-user tests under various surroundings such as indoors/outdoors, lighting conditions, and shade/sunlight with various backgrounds (**Fig. 4e**, **Supplementary Fig. 4**, and **Supplementary Video 2**). Every user took three images via a smartphone app from the LFA tests of nine clinical samples (6 positives for COVID-19 and 3 healthy controls), and then acquired the results. The accuracy of the 135 app-based tests with different users/smartphones was determined as 98%.

Second, we validated the universality by testing an additional seven different LFA models (**Fig. 4f**, **Supplementary Fig. 5**, **Supplementary Table. 5**, and **Supplementary Video 3**). We used Panbio COVID-19 Ag, BIO CREDIT COVID-19 Ag, SGT-flex COVID-19 Ag, GENEDIA COVID-19, Humasis COVID-19 Ag Test, Genbody COVID-19 Ag, InstaView COVID-19. Note that we carried out no additional training, meaning that we used the SMART^{AI}-LFA algorithm trained with LFA model 1 (COVID-19 Ag LFA kits, Calth Inc.). Then, we validated SMART^{AI}-LFA using three LFA models (n=360, Rapigen, SD biosensor, and Yuhan, Republic of Korea, See **Supplementary Table. 5**) and finally tested seven different models with smartphone app-based image acquisition (n=9,450). Every LFA kit was tested with 1,350 data (900 positives and 450 negatives). To avoid overfitting during the learning process, we tried to validate the model using the different LFA models from different manufacturers used in learning and testing. The sensitivity and specificity of Model 1 (Calth Inc.) from the confusion matrix are 99.6 % and 99.3%, respectively. Interestingly, the average sensitivity and specificity with seven different LFA models from 9,450 app-based tests were 94.0% and 89.7%, respectively. The total averaged sensitivity and specificity from eight different LFA models (LFA model 1 and 7 different models) were determined as 94.8% and 90.9%, respectively, indicating good universality of SMART^{AI}-LFA.” (line 303-329)

Comment 2: The development of a hardware-free system versus a hardware-dependent system for LFA is not necessarily significant. Phone attachment can be more useful than having an

algorithm that is limited to the used type of LFA device, probably due to the use of object finding/recognition.

Answer) First, as mentioned in answer to comment 1, for smartphone assay, two main approaches are cradle and cradle-free. Phone attachment (cradle) could be one of the great approaches for smartphone-based diagnostics. However, there are several limitations to the use of external cradles. The main issue is the universality of external cradles. Each smartphone has a different size with different camera locations, so the “one cradle fits all” strategy are not work. One needs each cradle for each smartphone. Moreover, many external cradles have their CCD for reading light signal Bluetooth for transmitting a signal to smartphones, consequently increasing the costs. We revised the manuscript with citations in *line 88-98*.

Second, since the current number of smartphone users worldwide is 6.648 billion, meaning 83.07% of the world’s population owns a smartphone, we expect our SMART^{AI}-LFA can provide “equipment-free” detection via smartphone. We revised as *“Since the number of current smartphone users exceeds 6 billion, meaning >80% of the world's population owns smartphones (<https://www.statista.com/statistics/330695/number-of-smartphone-users-worldwide/>), the assay using a smartphone can give accessibility and affordability. Smartphone-based assays have been performed to detect sperm concentration and motility²⁷, protein biomarkers²⁸, CRISPR-read SARS-CoV-2²⁵, Zika virus²⁹, norovirus³⁰, cell migration³¹, SARS-CoV-2 variants at single-nucleotide resolution³². To further improve the performance of smartphone assays, AI-assisted assays were performed for DNA diagnosis in malaria detection²⁴, HIV rapid tests³³, CRISPR-Cas13a based SARS-CoV-2 detection²⁶, and serological SARS-CoV-2 antibody test²¹.”* (line 64-73).

Third, we tried to show the universality by validating the ability of multi-users and multi-LFA models (fig.4g-h). *See line 303-329*. With newly added results, we expect our smartphone AI to be more useful than phone attachments as mentioned in comments 1.

Comment 3: The clinical use of this system is unclear and not well justified. Although, the reported accuracy is > 99%, what are the diagnostic recommendations after using this system? Is it significantly different than using other LFA systems? For instance, what is the recommended resolution when there is a conflict between a clinician and the AI-LFA system?

Answer) A recent paper in Nat Med (2021) claimed that the diagnostics of LFA with AI have the potential to provide a platform for workforce training, quality assurance, decision support, and mobile connectivity to inform disease control strategies, strengthen healthcare system efficiency and improve patient outcomes and outbreak management in emerging infections (Turbé, V. et al. Deep learning of HIV field-based rapid tests. Nat Med 27, 1165–1170 (2021)). Similarly, SMART^{AI}-LFA could provide a platform for workforce training, quality assurance, decision support, and mobile connectivity.

At the current stage, we expect that SMART^{AI}-LFA provides auxiliary functions of LFA, not replaceable PCR. Recently published papers in NEJM (New Engl J Med 383, e120 (2020) and BMJ (Bmj 372, n208 (2021)) claimed new diagnostic strategies for handling pandemics. Authors reported that the best Covid filter could be achieved using frequent, low-cost, simple, and rapid tests because SARS-CoV-2 quickly grows and spreads out exponentially. Therefore, rapid and frequent testing for COVID-19 is essential for minimizing virus transmission, especially before recognizing symptoms and in asymptomatic cases. I expect smartphone AI can help the strategy of NEJM and BMJ in the near future since it provides more accuracy and affordability with REASSURED criteria. Moreover, this smartphone AI platform can apply to semiquantitative POCT.

In case of conflict between the clinician and AI-LFA, we expect the clinician can make the final decision. As mentioned in the revised manuscript (*See the tunability of sensitivity and specificity, line 330-344*), we could control the sensitivity and specificity if necessary. The SMART^{AI}-LFA could benefit untrained users with its simplicity and accuracy since many non-experts have trouble analyzing LFA signals.

Comment 4: The study is missing a detailed comparison with a ‘gold standard’ technique currently used in diagnosing COVID-19, i.e., RT-qPCR or other NAT systems.

Answer) We collected all the clinical samples under the Institutional Review Board of Seoul St. Mary’s hospital (KC21TIDI0134K), then acquired Ct values for most of the clinical samples.

We revised as *“The information pertained to SARS-CoV-2 patients (n=45) and healthy controls (n=20) including sample collection, variants, sex, age, and Ct value (Supplementary Table. 3). All the samples were analyzed with RT-qPCR, then conducted the LFA assay. Finally, the positive data from LFA assay were classified into four groups, i.e., high/middle/middle-low/low titer, with the aid of the color chart level”* (line 189-195)

Specific weaknesses

Comment 1. Needs a major effort for editing towards improving the writing flow, simplified terminologies, and a smaller number of abbreviations.

Answer) As mentioned in previous answers, we tried to revise all the figures and manuscripts fully. Also, we improved the writing flow, terminologies, and abbreviations and corrected all the minor points, grammatical issues, and typos.

Comment 2. How your system performed at low virus titers? It can be useful to present these data in comparison to other tested algorithms and assays.

Answer) Our AI outperformed humans at low virus titers (*See revised fig.4h and line 345356*). We newly demonstrated the accuracy according to test data, including three different low-titer ratios (31, 61, and 91%), showing that the AI gave more accuracy than the humans without test set bias. See *“Finally, we showed the accuracy according to test data, including three different low-titer ratios (31, 61, and 91%) (Fig. 4h). We designed blind tests with 150 data (100 positives and 50 negatives) for each untrained individual (n=3) and human expert (n=3), then compared it with SMART^{AI}-LFA. We evenly distributed the tests (high/middle/middle-low) with low-titer ratios. For example, for a 31% low titer design, we prepared 23% high, 23% middle, 23% middle-low, and 31% low. Thus, we investigated the significant decrease in the accuracy of humans for lower concentrations, representing the human bias under lower concentrations. With an increase in the lower titer ratio (31 to 91%), we observed a significant decrease in the accuracy of untrained individuals (72.6 to 51.6%) and human experts (80.2 to 57.6%). Meanwhile, we noticed that the accuracy of SMART^{AI}-LFA is maintained at over 99.0% (100 to 99.3%). The accuracy data in Fig. 4h explicitly confirms the more reliable performance of SMART^{AI}-LFA in comparison to humans (untrained individuals and human experts).”*

Comment 3. Although the algorithm combining the two models sounds new, it presents a limitation to the developed system. The training and testing sets used for this study are focused on LFA devices produced by a single provider, without considering the variations that already exist in LFA systems' material, structure and dimensions. Other hardware-free systems are there, reporting a higher detection efficiency and sensitivity.

Answer) To address reviewer's concerns, we newly added the universality by validating the ability of multi-users and multi-LFA models (See newly added fig.4e-f) and manuscript as *“An important parameter of the AI-assist app is universality. We confirmed the universality by validating the ability of multi-users and multi-LFA models (Fig. 4e-f). First, five smartphone users with different smartphone models (LG Q51, Galaxy A52, iPhone 12 mini, iPhone 11 Pro, and iPhone 14 Max; Supplementary Table. 4) tested the smartphone app-based diagnostics for multi-user tests under various surroundings such as indoors/outdoors, lighting conditions, and shade/sunlight with various backgrounds (Fig. 4e, Supplementary Fig. 4, and Supplementary Video 2). Every user took three images via a smartphone app from the LFA tests of nine clinical samples (6 positives for COVID-19 and 3 healthy controls), and then acquired the results. The accuracy of the 135 app-based tests with different users/smartphones was determined as 98%.*

Second, we validated the universality by testing an additional seven different LFA models (Fig. 4f, Supplementary Fig. 5, Supplementary Table. 5, and Supplementary Video 3). We used Panbio COVID-19 Ag, BIO CREDIT COVID-19 Ag, SGT-flex COVID-19 Ag, GENEDIA COVID-19, Humasis COVID-19 Ag Test, Genbody COVID-19 Ag, InstaView COVID-19. Note that we carried out no additional training, meaning that we used the SMART^{AI}-LFA algorithm trained with LFA model 1 (COVID-19 Ag LFA kits, Calth Inc.). Then, we validated SMART^{AI}-LFA using three LFA models (n=360, Rapigen, SD biosensor, and Yuhan, Republic of Korea, See Supplementary Table. 5) and finally tested seven different models with smartphone app-based image acquisition (n=9,450). Every LFA kit was tested with 1,350 data (900 positives and 450 negatives). To avoid overfitting during the learning process, we tried to validate the model using the different LFA models from different manufacturers used in learning and testing. The sensitivity and specificity of Model 1 (Calth Inc.) from the confusion matrix are 99.6 % and 99.3%, respectively. Interestingly, the average sensitivity and specificity with seven different LFA models from 9,450 app-based tests were 94.0% and 89.7%, respectively. The total averaged sensitivity and specificity from eight different LFA models (LFA model 1 and 7 different models) were determined as 94.8% and 90.9%, respectively, indicating good universality of SMART^{AI}-LFA.” (line 303-329)

Fig. 4. Clinical tests with a smartphone application. (e-f) Universality test. (e) The accuracy of the 135 app-based tests with different users/smartphones, showing 98% accuracy. **(f)** The total averaged sensitivity and specificity from eight different LFA models (LFA model 1 and 7 different models) were determined as 94.8% and 90.9%, respectively.

Comment 4. Need to define what is a standard sample at the first time mentioned in the manuscript.

Answer) we define as “We train the algorithm with SARS-CoV-2 nucleocapsid protein spiked into phosphate buffered saline (PBS) buffer (referred as standard data)” (line 77-78) in the first time mentioned. Also, we showed the data preparation process for SMART^{AI}-LFA in Supplementary Fig. 1. (See revised Fig.S1)

Comment 5. What was the ‘gold standard’ method used to perform the blind test, was the PCR or target spiked samples? Would this test eliminate the need for PCR? If a test result was not approved by an expert human, what would be its significance from the healthcare point of view? Answer) The gold standard of blind tests is RT-qPCR of clinical samples. In training, we used standard samples and clinical samples. But, for the blind test, we tested only clinical samples labeled with RT-qPCR. We revised as “We assess the blind test using three different groups: untrained individuals, human experts, and SMART^{AI}-LFA using 1,500 test images (1,000 positives and 500 negatives). We collected clinical samples from COVID-19 patients at Seoul

St. Mary's Hospital. The information pertained to SARS-CoV-2 patients (n=45) and healthy controls (n=20) including sample collection, variants, sex, ages, and Ct values (Supplementary Table. 3). All the samples were analyzed with RT-qPCR, then conducted the LFA assay. Finally, the positive data from LFA assay were classified into four groups, i.e., high/middle/middle-low/low titer, with the aid of the color chart level (high with levels 10–8, middle with level 7–5, middle-low with level 4–3, and low titer with level 2–1 for positive, and negative with level 0). For positive data (n=1,000), we evenly distributed data across four groups (n=250). We prepared negative data (health controls, n=500). For the blind test, ten untrained individuals and ten human experts tested each of the 150 test images, which included 25 high, 25 middle, 25 middle-low, 25 low, and 50 negative data. The sum of the blind tests for both untrained individuals and human experts was 1500 images.” (line 187-200)

As mentioned in comment 3, this smartphone AI cannot replace PCR since the current commercialized LFA test kits have limited performance with lower sensitivity with larger CV than molecular diagnostics. Our paper titled "PCR-like Performance of Rapid Test with Permselective Tunable Nanotrap" is under revision in Nature Communications (2022). If one can increase the performance of the LFA like Nanotrap paper, the combination of smartphone AI and highly sensitive LFA could give a chance of daily monitoring, as claimed in two recent papers (New Engl J Med 383, e120 and Bmj 372, n208).

Regarding the concerns of "If a test result was not approved by an expert human" we think it depends on the performance optimization. We newly added tunability of sensitivity and specificity via the training data selection (*See Fig. 4g, line 330-344*). We expect we could control the sensitivity and specificity when we need no outperform of AI.

Comment 6. The advantage of developing a cradle-free smartphone system is debatable for LFA. LFA device might need optic-free and inexpensive attachment for accurate imaging. I would revise the statement in Line 83.

Answer) As the reviewer mentioned, the LFA device might need an optic-free and inexpensive attachment for accurate imaging. However, for higher performance, currently, many external cradles have their CCD for reading light signal Bluetooth for transmitting a signal to smartphones, consequently increasing the costs.

First, as mentioned in answer to comment 1 and 2, we revised the manuscript with citations in *line 88-98*. Second, since the current number of smartphone users worldwide is 6.648 billion,

meaning 83.07% of the world's population owns a smartphone, we expect our SMART^{AI}-LFA can provide “equipment-free” detection via smartphone. We revised it in *line 64-73*. Third, we tried to show the universality by validating the ability of multi-users and multi-LFA models (fig.4g-h). With newly added results, we expect our smartphone AI to be more useful than phone attachments as mentioned in comments 1 and 2 and 6. *See line 303-329*.

Comment 7. P5 L98: Untrained individuals use the NOVICA app to obtain the test results; However, the decision is made by human experts, not by AI. Why do authors consider this as a drawback or limitation? Would AI be solely used for deciding and recommending treatment?

Answer) When untrained individuals have difficulties interpreting the kit's result, Abbott's NAVICA App could be a great solution. However, the needs of experts need extra costs, which reduces the advantage of LFA's affordability. According to the reviewer's comments, we determined to delete the sentences for the NAVICA app. as *"Abbott developed the NOVICA application to pair with their rapid antigen test for COVID 19. Untrained individuals use the NOVICA app to obtain the test results; However, the decision is made by human experts, not by AI."*

Comment 8. P5 L104, the blind test was performed with n = 82, but only 62 samples are reported in the paragraph. Please check and correct it to reflect the right number.

Answer) We checked/corrected all the sample numbers in the revised manuscript. We showed the all the information in supplementary tables. (See *Supplementary Table 1 to 5*)

Supplementary Table 1. Training datasets of standard and patient samples for algorithm development.

Supplementary Table 2. Test and validation datasets of patient samples for testing and evaluating the algorithm. We carried out all the clinical tests using patient samples in test and validation.

Supplementary Table 3. SARS-CoV-2 patients (n=45) and healthy controls (n=20) information including sample collection, variants, sex, ages, and Ct values.

Supplementary Table 4. Multi-users and multi-smartphone models for validating universality for the tests of six patients and three healthy controls (normal).

Supplementary Table 5. Universality of different LFA kits. Validation with the LFAs of three different manufacturers and test kits from seven different manufacturers.

Comment 9. The blind test or other clinical tests were performed for pre-diagnosed or post-diagnosed individuals/samples, with and without symptoms. Any more information about the virus titer?

Answer) Most clinical samples are post-diagnosed with symptoms. All the samples have Ct values, and we added patients' information (*See Supplementary Table. 3*).

Supplementary Table 3. SARS-CoV-2 patients (n=45) and healthy controls (n=20) information including sample collection, variants, sex, ages, and Ct values.

Patient number	Sample collections	Variants	Sex (0: female 1: male)	Ages	Ct values
P#1	Saliva	Omicron	1	44	22.2
P#2	Saliva	Omicron	0	53	20.65
P#3	Saliva	Omicron	0	73	28.12
P#4	Saliva	Omicron	1	64	28.25
P#5	Saliva	Omicron	1	38	26.84
P#6	Saliva	Omicron	0	33	22.84
P#7	Saliva	Omicron	1	53	22.52
P#8	Saliva	Omicron	0	57	31.72
P#9	Saliva	Omicron	1	48	31.22
P#10	Saliva	Omicron	0	62	30.65
P#11	Saliva	Omicron	0	25	22.22
P#12	Saliva	Omicron	0	53	26.84
P#13	NP/OP	Omicron	-	-	21.44
P#14	NP/OP	Omicron	-	-	17.8
P#15	NP/OP	Omicron	1	44	18.72
P#16	NP/OP	Omicron	1	83	23.48
P#17	NP/OP	Omicron	0	33	22.84
P#18	NP/OP	Omicron	1	65	20.94
P#19	NP/OP	Omicron	0	39	22.7

P#20	NP/OP	Omicron	1	65	17.81
P#21	NP/OP	Omicron	0	25	22.22
P#22	NP/OP	Omicron	0	62	17.8
P#23	NP/OP	Omicron	0	53	26.84
P#24	NP/OP	Omicron	1	46	26.78
P#25	NP/OP	Omicron	0	78	28.12
P#26	NP/OP	Omicron	1	58	26.5
P#27	NP/OP	Omicron	1	46	25.69
P#28	NP/OP	Omicron	1	38	26.84
P#29	NP/OP	Omicron	1	64	28.25
P#30	NP/OP	Omicron	1	55	23.9
P#31	NP/OP	Omicron	1	22	26.78
P#32	NP/OP	Omicron	1	41	22.32
P#33	NP/OP	Omicron	1	41	22.32
P#34	NP/OP	Omicron	0	36	22.84
P#35	NP/OP	Delta	1	16	32
P#36	NP/OP	Delta	1	18	31
P#37	NP/OP	Delta	1	19	29
P#38	NP/OP	Delta	1	18	29
P#39	NP/OP	Delta	0	19	28
P#40	NP/OP	Delta	0	35	23.7
P#41	NP/OP	Delta	0	42	20.9
P#42	NP/OP	Delta	0	55	19.2
P#43	NP/OP	Delta	1	59	24.9
P#44	NP/OP	Delta	0	63	24.1
P#45	NP/OP	Omicron	1	27	Daily progress

healthy control	Sample type	Variant	Sex	Ages	Ct value
			(0: female 1: male)		
N#1	NP/OP	-	1	34	N

N#2	NP/OP	-	26	N
N#3	NP/OP	-	27	N
N#4	NP/OP	-	38	N
N#5	NP/OP	-	27	N
N#6	NP/OP	-	-	N
N#7	NP/OP	-	27	N
N#8	NP/OP	-	34	N
N#9	NP/OP	-	31	N
N#10	NP/OP	-	34	N
N#11	NP/OP	-	27	N
N#12	NP/OP	-	26	N
N#13	NP/OP	-	23	N
N#14	NP/OP	-	23	N
N#15	NP/OP	-	26	N
N#16	NP/OP	-	-	N
N#17	NP/OP	-	26	N
N#18	NP/OP	-	27	N
N#19	NP/OP	-	-	N
N#20	NP/OP	-	27	N

Comment 10. Classification or classifications?

Answer) Classification is generally used in the singular form. We corrected it.

Comment 11. P4 L78, 'Results' should be 'Results and Discussions.'

Answer) We corrected it.

Comment 12. P5 L115, define what are 'real measuring conditions.' This statement is unclear: several undefined terms are used e.g., real diagnostics, severe and critical cases, and the test conditions.

Answer) We changed all the undefined terms from as "*Many existing methods do not work well under different surroundings without external cradles because diagnostic surroundings include cases such as indoors/outdoors, lighting conditions, and shade/sunlight^{25,36}.*" (line 125127).

Comment 13. Throughout the manuscript, authors shift from using the term sample to the term data. It is highly recommended that authors mention the actual number of samples and the extracted data points used for training, testing and validation sets.

Answer) We carefully checked the number of samples and data in the revised manuscript, figures, and supplementary data. We first added Supplementary Fig. 1, which depicted data preparation for SMART^{AI}-LFA from standard samples/ clinical samples to training/validation/test dataset. The main corrections are "*Clinical samples (n=65, COVID-19 patients: 45 and healthy controls: 20) were tested to validate the clinical predictability of SMART^{AI}-LFA. (line 118-120). "We trained 8,914 (positive: 5,801 and negative: 3,113) for the first dataset (1st data set: standard data)." (line 436-437) " For the second dataset, we used clinical samples and prepared 8,005 COVID-19 training data (5,667 positives and 2,338 negatives) and 1,458 test data (1,026 positives and 432 negatives)." (line 437-439). Please check the revised *Supplementary Table 1 to 5.**

Comment 14. P6 L125, 126, what are the standard training data, and the mentioned number of 8,914 represents samples or data points, and how many samples are used?

Answer) As mentioned in comment 13, we corrected/revised the samples/data.

Comment 15. Untrained individuals (59.5–100%) and human experts (79.5–99.3%), this difference in the results of ROC analysis can indicate bias in the data set used for this comparative testing. Detailed data are needed.

Answer) As mentioned, to address bias issues, we fully redesigned the blind test starting from the unbiased data. For this, we increased the test set number from 80 to 1,500, then evenly

distributed the data set to minimize the bias of test results. *(See line 117-121, line 187-215, fig. Fig. 3. (a-e) Clinical validation via blind tests).*

With an increase in the lower titer ratio (31 to 91%), we observed a significant decrease in the accuracy of untrained individuals (72.6 to 51.6%) and human experts (80.2 to 57.6%). Meanwhile, we noticed that the accuracy of SMART^{AI}-LFA is maintained at over 99.0% (100 to 99.3%). The accuracy data in Fig. 4h explicitly confirms the more reliable (less biased) performance of SMART^{AI}-LFA in comparison to humans (untrained individuals and human experts). *(See line 345-356 and fig. 4h).*

Comment 16. P9 L199, was the LOD estimated using the same set of LFA devices/samples?
Answer) We determined the limit of detection (LOD) from the blind test using same images from LFA tests for untrained individuals, human experts, and SMART^{AI}-LFA. we revised as *“We tested the accuracy of the blind test using data sets (n = 1,500) from clinical samples. To calculate the LOD values, we prepared a second dataset (n=34: 26 positives and 8 negatives) with different concentrations, then set the LOD values under the manufacturer’s guidelines with 19/20 criteria, which represents the LOD is determined as the lowest concentration where 95% (19/20) are positive (<https://www.fda.gov/media/137302/download>).*”

Comment 17. The manuscript might benefit from a schematic diagram that presents the different steps and stages followed for app development.

Answer) We added schematic diagram in fig.4a-b and revised manuscript with supplementary video as *“Fig. 4a shows the smartphone applications (apps); we developed an AI-assisted assay based on smartphone apps using cloud-based deep learning (Supplementary Video 1). Since the distance between the camera of smartphone and LFA test kit might influence the image size and quality, we provided a guideline (blue line in Fig. 4a) for image capture processing. Following these guidelines, individual users can capture the images without considering the distance between the camera and the LFA test kit.*

Fig. 4b shows a schematic diagram depicting the data flows between smartphones and servers. The image of the test results acquired from a smartphone are sent to a server (Amazon Web Service, AWS) with image bytes, and the test results determined by the deep learning algorithm

are transmitted back in the JSON format to the smartphone applications. We imported the Flask module, a micro web framework written in Python, of SMART^{AI}-LFA and created a Flask web server on Amazon Cloud. CNN models with YOLOv3 and ResNet-18 are used to realize the sample-to-answer platform of the COVID-19 POCT. LFAs generally require assay times of up to 15 min. In app operation, the additional time needed from taking the image to returning the result is generally within tens of seconds, depending on the network and smartphones.” (line 259-273)

See the fig.4b as below. **“Fig. 4. Clinical tests with a smartphone application. (a) Smartphone application and (b) Schematic diagram depicting the data flows from a smartphone to the server (AWS), where the two algorithms are located.”**

Comment 18. What was the used RT-PCR kit, and any specific reason for using this specific kit?

Answer) We followed the In-house protocol used in Seoul St. Mary's Hospital. This protocol was mainly developed by the Institute Pasteur (Paris), one of the WHO reference laboratories. Based on this protocol, we used SuperScriptTM III Platinum[®] One-Step Quantitative RT-PCR System (Invitrogen) and LightCycler 480 real-time PCR machine (Roche). We revised as *“For RT-qPCR, we used an In-house protocol developed by the Institute Pasteur (Paris), one of the WHO reference laboratories. Based on this protocol, we used SuperScriptTM III Platinum[®] One-Step Quantitative RT-PCR System (Invitrogen) and LightCycler 480 real-time PCR machine (Roche).”* (line 410-414)

Comment 19. An invalid test can be because of the absence of the control line, but also due to the failure of reagents, and defective device, was this addressed during the development of the algorithm?

Answer) For the invalid test, we fully discussed it with the LFA manufacturers, then realized that most invalid case comes from the absence of the control line. One interesting case is the images without LFA kits. AI determined it as negative. With additional training, we can classify it as invalid. We expect that the case reviewer mentioned can be addressed by additional training in those cases. *See supplementary video 1 and 2.*

Comment 20. It would be useful to add a table that presents the data points or samples, and how each group (algorithm vs trained vs untrained) this will give a chance to for evaluating where the algorithm performed better. An image of the detection zone of the LFA device can also be very informative to analyze this performance.

Answer) To deliver clear information, we added new Supplementary Tables 1 to 5 with Supplementary Fig. 1 (*See Supplementary Table 1 to 5 and Supplementary Fig. 1*). Regarding an image of the detection zone, we can provide all the images with google drive links if the reviewer needs them. The size of images exceeds >500 GB; therefore, we showed one example of test line images in fig. 3d-e.

Example data used in this study is available at

<https://drive.google.com/file/d/16Rv9rcavScqK7UFFjZFcgys4vE3aZzn8/view?usp=sharing>

The overall source codes used in this study is available at https://github.com/Artinto/Sample-to-answer_COVID-19

Android's smartphone application (App) can be downloaded at

<https://drive.google.com/file/d/1zyp5I5q8dpqshWo1HhaTG8iBwfmqC6MI/view?usp=sharing>

Comment 21. The full name of AUC and its abbreviation should be fixed throughout the manuscript.

Answer) In the first appearance, we showed *“a larger value of the area under the curve (AUC) indicates a better classifier.”* (line 201-202).

Comment 22. It is unclear why the authors used DL v2 in clinical tests, and how the comparison with a more trained version of DLv3 is significant. A comparison should be performed versus the ‘gold standard’ technique, probably qPCR, as it is the currently used technique for diagnosing infection with SARS-CoV-2.

Answer) First, we renamed DL v2 and DL v3 and showed the comparison in fig.4c as *“In Fig. 4c, we illustrated the clinical training data effect on the ROC curves, representing accuracy. First, we depicted the ROC curves trained with standard data set (n=8,914, blue line, standard only), and then showed the enhanced ROC curves trained with additional clinical data (n = 8,005, red line, standard and clinical).”* (line 274-277). We also clarify all the sample preparation in new *Supplementary Tables 1 to 5 with Supplementary Fig. 1*. As we mentioned in previous answer, the gold standard of clinical samples is RT-qPCR and added patients' information in the supplementary table 3.

Comment 23. How do the authors explain that the SMART^{AI}-LFA app is over-performing xRcovid? The explanation provided is biased, considering that the app was developed with a single and specific LFA device, while xRcovid was developed using another LFA device. Again, providing a detailed table that presents data points is necessary for studies that focus on algorithm development. It will allow the audience to evaluate the developed algorithm and the best use of it.

Answer) First, we showed the universality by validating the ability of multi-users and multi-LFA models (*See newly added fig. 4e-h*). Second, xRcovid claimed their universality for other commercialized LFA kits. Since our SMART^{AI}-LFA and xRcovid claimed the universality, we determined to compare the power of the algorithm (*See revised fig. 4d*). We tried to show the effect of the algorithms and training effect as *“We represent the ROC curves and confusion matrix of the two algorithms (SMART^{AI}-LFA and xRcovid) from the clinical samples (Fig. 4d). The xRcovid is a published algorithm²¹ of artificial intelligence to improve COVID-19 rapid diagnostic tests and opened on Github ([HTTPS://github.com/dmendels-collab/xRcovid](https://github.com/dmendels-collab/xRcovid)) and Zenodo (https://zenodo.org/badge/latest/doi/31223_0700). To check the effect of training data and algorithm models, we prepared three cases: SMART^{AI}-LFA (our training set), xRcovid (our training set), and xRcovid. To assess the diagnostic accuracy of the models, we trained two*

models (SMART^{AI}-LFA and xRcovid) with our training set, which included the training data set from standard (n=8,914) and clinical data set (n = 8,005). Further, we compared it with the xRcovid algorithm with its pre-trained model²¹. We prepared 3,278 clinical tests (positive n=2,819 and negative n=459). The AUG reveals an enhanced accuracy (0.99) compared to that of xRcovid with our training set (0.94) and xRcovid (0.60) for clinical samples. The sensitivity and specificity of SMART^{AI}-LFA from the confusion matrix are 98.7% and 97.8%, respectively, which are notably higher than those of xRcovid with our training set (91.7% and 84.0%) and xRcovid (94.2% and 19.0%). The difference in AUG and accuracy between SMART^{AI}-LFA and xRcovid (our training set) indicates the algorithm's ability. Additionally, the larger differences in AUG and accuracy in xRcovid (our training set) and xRcovid indicate the effect of training data in clinical tests.” (line 285-302).

Comment 24. What is the ‘predicted level’ in Figure 4f, is it based on the sample line intensity? The numbering and the use of commas and hyphens in Figure captions need to be revised. Answer) We indicated ‘predicted level’ in fig. 3i (Figure 4f in original manuscript) from the fig.3g data. From the fig. 3g, we classified color intensity (0 to 12 levels) according to the nucleocapsid concentration (0 to 5 ng/ml). We newly added as *“The prediction level was based on the results of concentration prediction (Fig. 3g), classifying the prediction level (0 to 12) according to concentration (0 to 5 ng/ml).”* (line 248-250)

We want to thank you again for your valuable comments, and we found the comments very helpful in clarifying the originality of our work. We hope the manuscript, with careful revisions, meets your high standards.

Reviewer #2

Comments: Major revision

In this paper, the authors reported a deep learning-assisted smartphone-based LFA (SMART^{AI}-LFA) strategy for evaluation of COVID-19 via combining clinical data learning and a two-step CNN model (object findings and classification). With the new coronavirus raging around the world, how to use cheap test strips to get better accuracy does have great significance. However, from a technology development perspective, combining lateral flow analysis with AI, there are still many issues to be discussed. In addition, with all the validation of the technique and optimisation of the assay based on the technique, there are still a few comments that I would like to point out. In summary, I would recommend the article being considered for publication after the below concerns are addressed.

Answer) We really appreciate the reviewer's helpful comment. After carefully reading the reviewer's comments, we intensively revised all the figures, introduction, discussion, and conclusion parts with additional references.

First, we revised all the figures with additional experimental data. We revised fig.1 with simplified schemes and texts to clearly show the workflow of SMART^{AI}-LFA for COVID-19 diagnostics. We deleted the wordy text to deliver the main idea. In fig. 2, We redesigned/acquired the blind test with unbiased sample distribution (discussed later); then, we moved the data of (g) the blind test, (h) Cross-reactivity, and (i) the concentration prediction to revised fig.3. In revised fig.2, we focused only the algorithm optimization. In revised fig.3, we tried to show clinical validation via blind test, cross-reactivity, concentration prediction, and daily monitoring. We deleted the blind test of NP/OP and saliva (fig. 3(b-e) and fig. 3(f-i) in the original manuscript). Instead, we showed the blind test with increased test data from n=82 to n=1500 and an evenly distributed data set for acquiring unbiased results. Moreover, we revised fig.4 with newly added results. First, we showed the schemes of the smartphone app and data flows from a smartphone to the server (AWS) with new flow charts (See fig.4a-b). After showing the ROC curves of additional clinical samples onto standard samples (See fig.4c), we compared two algorithms (SMART^{AI}-LFA and xRcovid) (See fig.4d). We newly added the universality by validating the ability of multi-users and multi-LFA models (fig.4e-

f). Then, we show the effects of training data on sensitivity and specificity (fig.4g). Finally, we demonstrated the accuracy according to test data, showing a more reliable (unbiased) performance of SMART^{AI}-LFA than humans (fig.4h).

Second, we mainly revised the introduction parts as *“Although the real-time polymerase chain reaction (RT-qPCR) test is highly sensitive, frequent on-site tests of COVID-19 using RT-qPCR are still challenging. Generally, RT-qPCR can detect viral shedding up to 17 days after the infection period, increasing unnecessary quarantine^{2,3}. Moreover, long turnaround times of RT-qPCR allow infected people to spread the viruses exponentially before getting results.”* (See lines 47-51). and we added articles with smartphone AI as *“Since the number of current smartphone users exceeds 6 billion, meaning >80% of the world's population owns smartphones (<https://www.statista.com/statistics/330695/number-of-smartphone-users-worldwide/>), the assay using a smartphone can give accessibility and affordability. Smartphone-based assays have been performed to detect sperm concentration and motility²⁷, protein biomarkers²⁸, CRISPR-read SARS-CoV-2²⁵, Zika virus²⁹, norovirus³⁰, cell migration³¹, SARS-CoV-2 variants at single-nucleotide resolution³². To further improve the performance of smartphone assays, AI-assisted assays were performed for DNA diagnosis in malaria detection²⁴, HIV rapid tests³³, CRISPR-Cas13a based SARS-CoV-2 detection²⁶, and serological SARS-CoV-2 antibody test²¹.”* (See lines 64-73).

Third, we fully redesigned the blind test starting from the unbiased data. For this, we increased the test set number from 80 to 1,500, then evenly distributed the data set to minimize the bias of test results. We trained 8,005 images (positive: 5,667 and negative 2,338) and tested 1,500 images (positive: 1,000 and negative 500). All the blind test images are prepared from the COVID-19 patient/healthy control samples. We collected all the clinical samples from Seoul St. Mary's Hospital with Institutional Review Board Committee approval (KC21TIDI0134K) and acquired all the Ct values from the patient samples. The Ct value is not exactly inversely proportional to the color intensity of LFA test results; therefore, we prepared the blind tests set using the color chart under the LFA manufacturer's guidelines. Generally, the color chart consisted of 10 classifications; then, we evenly prepared the blind test images (high 250 images, mid 250 images, low 250 images, and very low 250 images) to prevent bias. We revised as follows: *“We determined the accuracy of the blind test using 1,500 test images from untrained individuals (n=10), human experts (n=10), and SMART^{AI}-LFA (Fig. 1b). Clinical samples*

($n=65$, COVID-19 patients: 45 and healthy controls: 20) were tested to validate the clinical predictability of SMART^{AI}-LFA. We arranged the training, test and validation datasets for algorithm development (**Supplementary Table 1-2, Supplementary Fig. 1**).” (line 117-121) and “We assess the blind test using three different groups: untrained individuals, human experts, and SMART^{AI}-LFA using 1,500 test images (1,000 positives and 500 negatives). We collected clinical samples of NP/OP samples from COVID-19 patients at Seoul St. Mary’s Hospital. The information pertained to SARS-CoV-2 patients ($n=45$) and healthy controls ($n=20$) including sample collection, variants, sex, age, and Ct value (Supplementary Table. 3). All the samples were analyzed with RT-qPCR, then conducted the LFA assay. Finally, the positive data from LFA assay were classified into four groups, i.e., high/middle/middle-low/low titer, with the aid of the color chart level (high with levels 10–8, middle with level 7–5, middle-low with level 4–3, and low titer with level 2–1 for positive, and negative with level 0). For positive data ($n=1,000$), we evenly distributed data across four groups ($n=250$). We prepared negative data (health controls, $n=500$). For the blind test, ten untrained individuals and ten human experts tested each of the 150 test images, which included 25 high, 25 middle, 25 middle-low, 25 low, and 50 negative data. The sum of the blind tests for both untrained individuals and human experts was 1500 images.

The ROC curve shows a general overview of the three different models; a larger value of the area under the curve (AUC) indicates a better classifier. From the ROC curve in Fig. 3b, we observe larger AUCs for SMART^{AI}-LFA (1.00) compared with that for untrained individuals (0.79) and human experts (0.86); this demonstrates that SMART^{AI}-LFA is an excellent classifier for clinical assays. Fig. 3c shows the table for the three different groups, which indicates a considerable enhancement in sensitivity and specificity using a SMART^{AI}-LFA (100 and 100%) compared with untrained individuals (72.9 and 86.0%) and human experts (83 and 88.2%). To closely study the reason for the accuracy reaching 100%, we explored the effect of the training and test datasets discussed later (Fig. 4g-h).

We present three positive clinical sample images in Fig. 3d to clarify the AI’s decision ability. Fig. 3e shows the evidence of a positive test line of Fig. 3d with contrast enhancement. Although all groups can provide the correct answer as seen from the first image, only SMART^{AI}-LFA can predict the positive samples from the third image, which confirms that deep learning-assisted SMART^{AI}-LFA can provide precision decisions superior to those of human experts.” (line 187-215)

Fourth, we relocated clinical validation of cross-reactivity, concentration prediction, and daily monitoring in fig. 3. (See revised fig.3 and manuscript parts).

Fifth, we fully revised fig. 4. We added *Supplementary Videos* to show the SMART^{AI}-LFA's operation. To depict the data flows between smartphones and servers (Fig. 4b), we revised as "**Fig. 4b** shows a schematic diagram depicting the data flows between smartphones and servers. The image of the test results acquired from a smartphone are sent to a server (Amazon Web Service, AWS) with image bytes, and the test results determined by the deep learning algorithm are transmitted back in the JSON format to the smartphone applications. We imported the Flask module, a micro web framework written in Python, of SMART^{AI}-LFA and created a Flask web server on Amazon Cloud. CNN models with YOLOv3 and ResNet-18 are used to realize the sample-to-answer platform of the COVID-19 POCT. LFAs generally require assay times of up to 15 min. In app operation, the additional time needed from taking the image to returning the result is generally within tens of seconds, depending on the network and smartphones." (line 265-273). To check the AI-assist app's universality, we first validated the ability of multi-users with multiple smartphones (Fig. 4e-f) as "An important parameter of the AI-assist app is universality. We confirmed the universality by validating the ability of multi-users and multi-LFA models (**Fig. 4e-f**). First, five smartphone users with different smartphone models (LG Q51, Galaxy A52, iPhone 12 mini, iPhone 11 Pro, and iPhone 14 Max; **Supplementary Table. 4**) tested the smartphone app-based diagnostics for multi-user tests under various surroundings such as indoors/outdoors, lighting conditions, and shade/sunlight with various backgrounds (**Fig. 4e, Supplementary Fig. 4, and Supplementary Video 2**). Every user took three images via a smartphone app from the LFA tests of nine clinical samples (6 positives for COVID-19 and 3 healthy controls), and then acquired the results. The accuracy of the 135 app-based tests with different users/smartphones was determined as 98%." (line 303-312). Then, we validated the universality by testing an additional seven different LFA models (Fig. 4f). "Second, we validated the universality by testing an additional seven different LFA models (**Fig. 4f, Supplementary Fig. 5, Supplementary Table. 5, and Supplementary Video 3**). We used Panbio COVID-19 Ag, BIO CREDIT COVID-19 Ag, SGT-flex COVID-19 Ag, GENEDIA COVID-19, Humasis COVID-19 Ag Test, Genbody COVID-19 Ag, InstaView COVID-19. Note that we carried out no additional training, meaning that we used the SMART^{AI}-LFA algorithm trained with LFA model 1 (COVID-19 Ag LFA kits, Calth Inc.). Then, we validated SMART^{AI}-LFA using three LFA models (n=360, Rapigen, SD biosensor, and

Yuhan, Republic of Korea, See **Supplementary Table. 5**) and finally tested seven different models with smartphone app-based image acquisition ($n=9,450$). Every LFA kit was tested with 1,350 data (900 positives and 450 negatives). To avoid overfitting during the learning process, we tried to validate the model using the different LFA models from different manufacturers used in learning and testing. The sensitivity and specificity of Model 1 (Calth Inc.) from the confusion matrix are 99.6 % and 99.3%, respectively. Interestingly, the average sensitivity and specificity with seven different LFA models from 9,450 app-based tests were 94% and 89.7%, respectively. The total averaged sensitivity and specificity from eight different LFA models (LFA model 1 and 7 different models) were determined as 94.8% and 90.9%, respectively, indicating good universality of SMART^{AI}-LFA.” (line 313-328). In fig. 4g, we tried to determine whether the training data is associated with the tunability of sensitivity and specificity as “The aim of **Fig. 4g** has been to identify the tunability of sensitivity and specificity. Generally, LFA manufacturers control the sensitivity and specificity of commercial LFA by optimizing chemistry, materials, and LFA design. To determine whether the training data is associated with the tunability of sensitivity and specificity, we controlled the ratio of the low titers in the training data. We trained the algorithm with the existing dataset (SMART^{AI}-LFA) and validated ($n=360$, Rapigen, SD biosensor, and Yuhan, Republic of Korea, See **Supplementary Table. 5**) and finally tested 10,908 data (7,326 positives and 3,582 negatives). We visualized the confusion matrix, which allows the extraction of the true negatives (TN), true positives (TP), false negatives (FN), and false positives (FP). The true positive and true negative portions of the confusion matrix indicate the sensitivity and specificity of the clinical assay, respectively. With more low titer (high/middle/middle-low/low titer=20/25/30/25%) of test data, we acquired higher sensitivity for eight different LFA models (LFA model 1 and 7 different models) as 94.8% with 90.9% specificity. Meanwhile, the algorithm trained without low titer samples (high/middle/middle-low/low titer=27/33/40/0 %) yielded higher specificity (99 %) with less sensitivity (86%).”. In fig. 4h, we tried to show the effects of test data on the accuracy. “Finally, we showed the accuracy according to test data, including three different low-titer ratios (31, 61, and 91%) (**Fig. 4h**). We designed blind tests with 150 data (100 positives and 50 negatives) for each untrained individual ($n=3$) and human expert ($n=3$), then compared it with SMART^{AI}-LFA. We evenly distributed the tests (high/middle/middle-low) with low-titer ratios. For example, for a 31% low titer design, we prepared 23% high, 23% middle, 23% middle-low, and 31% low. Thus, we investigated the significant decrease in the accuracy of humans for lower concentrations, representing the human bias under lower concentrations. With an increase in the lower titer ratio (31 to 91%), we observed a significant

decrease in the accuracy of untrained individuals (72.6 to 51.6%) and human experts (80.2 to 57.6%). Meanwhile, we noticed that the accuracy of SMART^{AI}-LFA is maintained at over 99% (100 to 99.3%). The accuracy data in Fig. 4h explicitly confirms the more reliable performance of SMART^{AI}-LFA in comparison to humans (untrained individuals and human experts).” (line 330-356).

Lastly, we added major limitation of SMART^{AI}-LFA at the end of “Results and Discussions” part as *“The SMART^{AI}-LFA study had a major limitation: image quality depends on the smartphone. the high-end smartphone has an automatic filter function, which decreases data accuracy because the smartphone automatically corrects its image quality. Therefore, further studies need to focus on acquiring raw data. Another limitation is that the accuracy could be reduced if the test surroundings are outside the training data. Testing the surrounding out-of-distribution training data potentially limits the accuracy of the current deep learning model. Retest signs in the image acquisition stage can be considered a solution to address these problems. Guided by the retest sign, images can be obtained within the training-data distribution.”* (line 357-364). Then, we added new sentences in the conclusion parts as *“(3) Validation of multi-users and multi-LFA models showed the universality of SMART^{AI}-LFA. Since smartphone users exceed 6 billion, SMART^{AI}-LFA with smartphone can improve the POCT along with affordability.”* (line 376-378)

We hope the manuscript, with careful revisions, meets your high standards. Below we provide the point-by-point responses. All modifications in the manuscript have been highlighted in red.

Comment 1: From a technology development perspective, with the aid of AI, the low-cost, user-friendly characteristics of lateral flow assays are weakened. After all, some poor areas/conditions do not have a smartphone/internet and can only detect the results by human eye observation. In other words, what is the significance brought about by the addition of AI to such a disposable rough testing technique? What are the implications of this for the development of future rapid tests? Please authors supplement the relevant viewpoint in introduction.

Answer) As the reviewer mentioned, some poor areas/conditions do not have a smartphone/internet and can only detect the results by human eye observation. However, since 83.07% of the world's population owns a smartphone (smartphone users worldwide are 6.648 billion), we expect the smartphone to meet the affordability and accessibility of LFA kits. I think these mobile user statistics could be the significance of SMART^{AI}-LFA. Moreover, the smartphone can give the REASSURED criteria (Real-time connectivity, Ease of specimen collection, Affordable, Sensitive, Specific, User friendly, Rapid and robust, Equipment free and Deliverable to end-users), which are the new criteria for digital connectivity (Nat Microbiol 4, 46-54).

We revised introduction part as *“Since the number of current smartphone users exceeds 6 billion, meaning >80% of the world's population owns smartphones (<https://www.statista.com/statistics/330695/number-of-smartphone-users-worldwide/>), the assay using a smartphone can give accessibility and affordability. Smartphone-based assays have been performed to detect sperm concentration and motility²⁷, protein biomarkers²⁸, CRISPR-read SARS-CoV-2²⁵, Zika virus²⁹, norovirus³⁰, cell migration³¹, SARS-CoV-2 variants at single-nucleotide resolution³². To further improve the performance of smartphone assays, AI-assisted assays were performed for DNA diagnosis in malaria detection²⁴, HIV rapid tests³³, CRISPR-Cas13a based SARS-CoV-2 detection²⁶, and serological SARS-CoV-2 antibody test²¹.”* (See lines 64-73).

For smartphone assay, two main approaches are cradle and cradle-free. Phone attachment (cradle) could be one of the great approaches for smartphone-based diagnostics. However, there are several limitations to the use of external cradles. The main issue is the universality of external cradles. Each smartphone has a different size with different camera locations, so the “one cradle fits all” strategy are not work. One needs each cradle for each smartphone. Moreover, many external cradles have their CCD for reading light signal Bluetooth for transmitting a signal to smartphones, consequently increasing the costs. We revised as *“The most significant advantage of SMART^{AI}-LFA is that it requires no external cradles attached to the smartphone. Since each smartphone has a different size and camera location, the one-cradle-fits-all approach may not work. Moreover, many external cradles need their external optics^{27, 31, 32, 34}, fluorescence components²⁵, microscope^{28, 30}, and Bluetooth^{24, 35} for reading and transmitting signals to smartphones, consequently increasing costs. Meanwhile, the use of AI could eliminate external cradles, which meets the universality of smartphone-based assays discussed later. Further, a smartphone-based AI provides a great opportunity to meet the*

REASSURED criteria (Real-time connectivity, Ease of specimen collection, Affordable, Sensitive, Specific, User friendly, Rapid and robust, Equipment free and Deliverable to end-users)²², which are the new criteria for digital connectivity.” (see line 88-98).

In conclusion parts, we revised as “With the availability of multi-users and multi-models with less bias for low titer samples, we expect that a SMART^{AI}-LFA will provide the necessary breakthroughs in early detection without reading ambiguities as in AI-assisted smartphone diagnostics.” (see line 388-391).

Comment 2: Is visual observation bias the biggest cause of false-positive and false negative problems in lateral flow assays? In this work, false negatives due to visual bias are greatly reduced by AI image processing, so how is the problem of false positives resolved by AI? Please give the relevant discussion in paper.

Answer) We think the bias can be mainly from (1) the LFA kit's performance and (2) observation. To overcome the LFA kit's performance bias (false negative), which causes lower sensitivity, many researchers tried to develop highly sensitive POCT. For highly sensitive POCTs, we suggested the "PCR-like Performance of Rapid Test with Permselective Tunable Nanotrap" which is under revision in Nature Communications (2022). If commercialized, we expect it can address performance bias.

For observation bias, current self-testing for COVID-19 detection suffers from low accuracy and reading ambiguities attributable to highly subjective readings performed by untrained individuals. Especially, untrained individuals showed less sensitivity than experts. As shown in the blind test (*See revised fig.3a-e*), the untrained individuals showed lower sensitivity than the experts, meaning the bias between persons.

To address the problems of false positives, we newly added *fig. 4g and fig.4h* as “*The aim of **Fig. 4g** has been to identify the tunability of sensitivity and specificity. Generally, LFA manufacturers control the sensitivity and specificity of commercial LFA by optimizing chemistry, materials, and LFA design. To determine whether the training data is associated with the tunability of sensitivity and specificity, we controlled the ratio of the low titers in the training data. We trained the algorithm with the existing dataset (SMART^{AI}-LFA) and validated (n=360, Rapigen, SD biosensor, and Yuhan, Republic of Korea, See **Supplementary Table. 5**) and finally tested 10,908 data (7,326 positives and 3,582 negatives). We visualized the confusion matrix, which allows the extraction of the true negatives (TN), true positives (TP),*

false negatives (FN), and false positives (FP). The true positive and true negative portions of the confusion matrix indicate the sensitivity and specificity of the clinical assay, respectively. With more low titer (high/middle/middle-low/low titer=20/25/30/25%) of test data, we acquired higher sensitivity for eight different LFA models (LFA model 1 and 7 different models) as 94.8% with 90.9% specificity. Meanwhile, the algorithm trained without low titer samples (high/middle/middle-low/low titer=27/33/40/0%) yielded higher specificity (99.0%) with less sensitivity (86.0%).

Finally, we showed the accuracy according to test data, including three different low-titer ratios (3%, 6%, and 9%) (Fig. 4h). We designed blind tests with 150 data (100 positives and 50 negatives) for each untrained individual (n=3) and human expert (n=3), then compared it with SMART^{AI}-LFA. We evenly distributed the tests (high/middle/middle-low) with low-titer ratios. For example, for a 3% low titer design, we prepared 23% high, 23% middle, 23% middle-low, and 3% low. Thus, we investigated the significant decrease in the accuracy of humans for lower concentrations, representing the human bias under lower concentrations. With an increase in the lower titer ratio (3% to 9%), we observed a significant decrease in the accuracy of untrained individuals (72.6 to 51.6%) and human experts (80.2 to 57.6%). Meanwhile, we noticed that the accuracy of SMART^{AI}-LFA is maintained at over 99.0% (100 to 99.3%). The accuracy data in Fig. 4h explicitly confirms the more reliable performance of SMART^{AI}-LFA in comparison to humans (untrained individuals and human experts).” (line 330-356). We expect the data suggested above could address the reviewer’s concerns.

Comment 3: In result section, the authors stated that “We demonstrated the accuracy of the blind test (n = 82) using untrained individuals, human experts, and SMART^{AI}-LFA. Clinical samples (n=64, COVID-19 patients:42 and healthy controls: 20) that included nasopharyngeal/oropharyngeal (NP/OP) swabs and saliva samples” I think the authors should provide a table to show the SMART^{AI}-LFA’s individual accuracy of the blind test of Clinical samples (n=64, COVID-19 patients 42 and healthy 20; and healthy controls: 20).

Answer) We fully redesigned the blind test starting from the unbiased data. For this, we increased the test set number from 80 to 1,500, then evenly distributed the data set to minimize the bias of test results. We trained 8,005 images (positive: 5,667 and negative 2,338) and tested 1,500 images (positive: 1,000 and negative 500). All the blind test images are prepared from the COVID-19 patient/healthy control samples. We collected all the clinical samples from Seoul St. Mary’s Hospital with Institutional Review Board Committee approval

(KC21TIDI0134K) and acquired all the Ct values from the patient samples. The Ct value is not exactly inversely proportional to the color intensity of LFA test results; therefore, we prepared the blind tests set using the color chart under the LFA manufacturer's guidelines. Generally, the color chart consisted of 10 classifications; then, we evenly prepared the blind test images (high 250 images, mid 250 images, low 250 images, and very low 250 images) to prevent bias. We revised as follows: *“We determined the accuracy of the blind test using 1,500 test images from untrained individuals (n=10), human experts (n=10), and SMART^{AI}-LFA (Fig. 1b). Clinical samples (n=65, COVID-19 patients: 45 and healthy controls: 20) were tested to validate the clinical predictability of SMART^{AI}-LFA. We arranged the training, test and validation datasets for algorithm development (Supplementary Table 1-2, Supplementary Fig. 1).”* (line 117-121).

Then, we added blind test results as *“We assess the blind test using three different groups: untrained individuals, human experts, and SMART^{AI}-LFA using 1,500 test images (1,000 positives and 500 negatives). We collected clinical samples of NP/OP samples from COVID-19 patients at Seoul St. Mary’s Hospital. The information pertained to SARS-CoV-2 patients (n=45) and healthy controls (n=20) including sample collection, variants, sex, age, and Ct value (Supplementary Table. 3). All the samples were analyzed with RT-qPCR, then conducted the LFA assay. Finally, the positive data from LFA assay were classified into four groups, i.e., high/middle/middle-low/low titer, with the aid of the color chart level (high with levels 10–8, middle with level 7–5, middle-low with level 4–3, and low titer with level 2–1 for positive, and negative with level 0). For positive data (n=1,000), we evenly distributed data across four groups (n=250). We prepared negative data (health controls, n=500). For the blind test, ten untrained individuals and ten human experts tested each of the 150 test images, which included 25 high, 25 middle, 25 middle-low, 25 low, and 50 negative data. The sum of the blind tests for both untrained individuals and human experts was 1500 images.*

The ROC curve shows a general overview of the three different models; a larger value of the area under the curve (AUC) indicates a better classifier. From the ROC curve in Fig. 3b, we observe larger AUCs for SMART^{AI}-LFA (1.00) compared with that for untrained individuals (0.79) and human experts (0.86); this demonstrates that SMART^{AI}-LFA is an excellent classifier for clinical assays. Fig. 3c shows the table for the three different groups, which indicates a considerable enhancement in sensitivity and specificity using a SMART^{AI}-LFA (100 and 100%) compared with untrained individuals (72.9 and 86.0%) and human experts

(83 and 88.2%). To closely study the reason for the accuracy reaching 100%, we explored the effect of the training and test datasets discussed later (Fig. 4g-h).

We present three positive clinical sample images in Fig. 3d to clarify the AI's decision ability. Fig. 3e shows the evidence of a positive test line of Fig. 3d with contrast enhancement. Although all groups can provide the correct answer as seen from the first image, only SMART^{AI}-LFA can predict the positive samples from the third image, which confirms that deep learning-assisted SMART^{AI}-LFA can provide precision decisions superior to those of human experts." (line 187-215).

Finally, we added Supplementary Fig. 1, which depicted data preparation for SMART^{AI}-LFA from standard samples/ clinical samples to training/validation/test dataset. The main corrections are "Clinical samples (n=65, COVID-19 patients: 45 and healthy controls: 20) were tested to validate the clinical predictability of SMART^{AI}-LFA. (line 118-120). "We trained 8,914 (positive: 5,801 and negative: 3,113) for the first dataset (1st data set: standard data)." (line 436-437) " For the second dataset, we used clinical samples and prepared 8,005 COVID-19 training data (5,667 positives and 2,338 negatives) and 1,458 test data (1,026 positives and 432 negatives)." (line 437-439). According to reviewer's suggestion, we newly added Supplementary Table 1 to 5 (See revised supplementary). I hope those blind test results with figures and tables address the reviewer's comments.

Supplementary Table 1. Training datasets of standard and patient samples for algorithm development.

Supplementary Table 2. Test and validation datasets of patient samples for testing and evaluating the algorithm. We carried out all the clinical tests using patient samples in test and validation.

Supplementary Table 3. SARS-CoV-2 patients (n=45) and healthy controls (n=20) information including sample collection, variants, sex, ages, and Ct values.

Supplementary Table 4. Multi-users and multi-smartphone models for validating universality for the tests of six patients and three healthy controls (normal).

Supplementary Table 5. Universality of different LFA kits. Validation with the LFAs of three different manufacturers and test kits from seven different manufacturers.

Comment 4: Commercial COVID-19 Ag LFA kits from different supplier have different the degree of coloring after loading COVID-19 sample solution. In order to verify the general applicability of the developed method, I suggest the authors choose LFA kits from more suppliers for deep learning process. Comment 5: In order to show the method's reliability, as

known, different people using different smartphone and taking pictures in different environments. I suggest the authors should collect more than 100 different person's pictures using their own smartphone then using SMART^{AI}-LFA to make corresponding analysis, and compared with their clinical COVID-19 results, and provide the testing accuracy that will be very meaningful.

Answer) Thanks for your helpful comments. As the reviewer suggested in comments 4&5, the important parameter of the AI-assist app is universality (multi-users and multi-LFA models). To answer comment 4, we showed the ability of multi-LFA models. To address comment 5, we tested multi-users/smartphone models. We newly added fig. 4e-f and added paragraphs as *“An important parameter of the AI-assist app is universality. We confirmed the universality by validating the ability of multi-users and multi-LFA models (Fig. 4e-f). First, five smartphone users with different smartphone models (LG Q51, Galaxy A52, iPhone 12 mini, iPhone 11 Pro, and iPhone 14 Max; Supplementary Table. 4) tested the smartphone app-based diagnostics for multi-user tests under various surroundings such as indoors/outdoors, lighting conditions, and shade/sunlight with various backgrounds (Fig. 4e, Supplementary Fig. 4, and Supplementary Video 2). Every user took three images via a smartphone app from the LFA tests of nine clinical samples (6 positives for COVID-19 and 3 healthy controls), and then acquired the results. The accuracy of the 135 app-based tests with different users/smartphones was determined as 98%.*

Second, we validated the universality by testing an additional seven different LFA models (Fig. 4f, Supplementary Fig. 5, Supplementary Table. 5, and Supplementary Video 3). We used Panbio COVID-19 Ag, BIO CREDIT COVID-19 Ag, SGT-flex COVID-19 Ag, GENEDIA COVID-19, Humasis COVID-19 Ag Test, Genbody COVID-19 Ag, InstaView COVID-19. Note that we carried out no additional training, meaning that we used the SMART^{AI}-LFA algorithm trained with LFA model 1 (COVID-19 Ag LFA kits, Calth Inc.). Then, we validated SMART^{AI}-LFA using three LFA models (n=360, RapiGen, SD biosensor, and Yuhan, Republic of Korea, See Supplementary Table. 5) and finally tested seven different models with smartphone app-based image acquisition (n=9,450). Every LFA kit was tested with 1,350 data (900 positives and 450 negatives). To avoid overfitting during the learning process, we tried to validate the model using the different LFA models from different manufacturers used in learning and testing. The sensitivity and specificity of Model 1 (Calth Inc.) from the confusion matrix are 99.6 % and 99.3%, respectively. Interestingly, the average sensitivity and specificity with seven different LFA models from 9,450 app-based tests were 94.0% and 89.7%, respectively. The

total averaged sensitivity and specificity from eight different LFA models (LFA model 1 and 7 different models) were determined as 94.8% and 90.9%, respectively, indicating good universality of SMART^{AI}-LFA.

The aim of **Fig. 4g** has been to identify the tunability of sensitivity and specificity. Generally, LFA manufacturers control the sensitivity and specificity of commercial LFA by optimizing chemistry, materials, and LFA design. To determine whether the training data is associated with the tunability of sensitivity and specificity, we controlled the ratio of the low titers in the training data. We trained the algorithm with the existing dataset (SMART^{AI}-LFA) and validated ($n=360$, Rapigen, SD biosensor, and Yuhan, Republic of Korea, See **Supplementary Table. 5**) and finally tested 10,908 data (7,326 positives and 3,582 negatives). We visualized the confusion matrix, which allows the extraction of the true negatives (TN), true positives (TP), false negatives (FN), and false positives (FP). The true positive and true negative portions of the confusion matrix indicate the sensitivity and specificity of the clinical assay, respectively. With more low titer (high/middle/middle-low/low titer=20/25/30/25%) of test data, we acquired higher sensitivity for eight different LFA models (LFA model 1 and 7 different models) as 94.8% with 90.9% specificity. Meanwhile, the algorithm trained without low titer samples (high/middle/middle-low/low titer=27/33/40/0%) yielded higher specificity (99.0%) with less sensitivity (86.0%).

Finally, we showed the accuracy according to test data, including three different low-titer ratios (31, 61, and 91%) (**Fig. 4h**). We designed blind tests with 150 data (100 positives and 50 negatives) for each untrained individual ($n=3$) and human expert ($n=3$), then compared it with SMART^{AI}-LFA. We evenly distributed the tests (high/middle/middle-low) with low-titer ratios. For example, for a 31% low titer design, we prepared 23% high, 23% middle, 23% middle-low, and 31% low. Thus, we investigated the significant decrease in the accuracy of humans for lower concentrations, representing the human bias under lower concentrations. With an increase in the lower titer ratio (31 to 91%), we observed a significant decrease in the accuracy of untrained individuals (72.6 to 51.6%) and human experts (80.2 to 57.6%). Meanwhile, we noticed that the accuracy of SMART^{AI}-LFA is maintained at over 99.0% (100 to 99.3%). The accuracy data in **Fig. 4h** explicitly confirms the more reliable performance of SMART^{AI}-LFA in comparison to humans (untrained individuals and human experts).” (line 303-356).

Fig. 4. Clinical tests with a smartphone application. (a) Smartphone application and (b) Schematic diagram depicting the data flows from a smartphone to the server (AWS), where the two algorithms are located. (c) The ROC curves according to training data (standard only (n=8,914) and additional clinical data (n = 8,005)). (d) The ROC curves and confusion matrix of the two algorithms (SMART^{AI}-LFA and xRcovid) for clinical tests (n = 3,278). (e-f) Universality test. (e) The accuracy of the 135 app-based tests with different users/smartphones, showing 98% accuracy. (f) The total averaged sensitivity and specificity from eight different LFA models (LFA model 1 and 7 different models) were determined as 94.8% and 90.9%, respectively. (g) The tunability of sensitivity and specificity according to the training data. (h) The accuracy according to test data, including three different low-titer ratios (31, 61, and 91%), showing more reliable performance of SMART^{AI}-LFA than humans.

Comment 6: What's the connection of YOLOv3, RESNET-18 and dl.v3 should be explain clearly. What improvements between versions should be further elaborated.

Answer) To avoid confusing expressions, we fully revised the manuscript. Please see the whole revised manuscript in red. We deleted some abbreviations like DL.v3 throughout the manuscripts. YOLOv3 is the algorithm for cropping the kit's test line, and ResNet-18 is the classification model. The DL.v2 is the model trained with standard samples; we renamed it as standard only (See fig.4c). In the same manner, we renamed DL.v3 (final trained model), as SMART^{AI}-LFA with additional training of clinical samples. One example is *"In Fig. 4c, we illustrated the clinical training data effect on the ROC curves, representing accuracy. First, we depicted the ROC curves trained with standard data set (n=8,914, blue line, standard only), and then showed the enhanced ROC curves trained with additional clinical data (n = 8,005, red line, standard and clinical)."* (line 274-277).

Comment 7: About cloud-based AI (AWS), I can't find where to import COVID-19 photos. Is it free or paid? If the authors can provide a webpage entrance of SMART^{AI}-LFA to let people make a test that would be more convincing and meaningful.

Answer) We paid the AWS service fee, but the users can use the service for free. Simply speaking, one can use a mobile application to acquire a test image, then the image is sent to the server where the AI algorithm is located. Finally, the server returned results (decisions) to each smartphone. To provide test webpages, we need additional development. Instead, we can provide the application source (Android & IOS). We provided links below. I hope it will give readers a more clear and more meaningful understanding.

To depict the data flows between smartphones and servers (Fig. 4b), we also revised as *"Fig. 4b shows a schematic diagram depicting the data flows between smartphones and servers. The image of the test results acquired from a smartphone are sent to a server (Amazon Web Service, AWS) with image bytes, and the test results determined by the deep learning algorithm are transmitted back in the JSON format to the smartphone applications. We imported the Flask module, a micro web framework written in Python, of SMART^{AI}-LFA and created a Flask web server on Amazon Cloud. CNN models with YOLOv3 and ResNet-18 are used to realize the sample-to-answer platform of the COVID-19 POCT. LFAs generally require assay times of up to 15 min. In app operation, the additional time needed from taking the image to returning the result is generally within tens of seconds, depending on the network and smartphones."* (line 265-273 and fig.4b).

Fig. 4. Clinical tests with a smartphone application. (b) Schematic diagram depicting the data flows from a smartphone to the server (AWS), where the two algorithms are located.

Example data used in this study is available at

<https://drive.google.com/file/d/16Rv9rcavScqK7UFFjZFCgys4vE3aZzn8/view?usp=sharing>

The overall source codes used in this study is available at https://github.com/Artinto/Sample-to-answer_COVID-19

Android's smartphone application (App) can be downloaded at

<https://drive.google.com/file/d/1zyp5I5q8dpqshWo1HhaTG8iBwfmqC6MI/view?usp=sharing>

Install manual is available at

https://drive.google.com/drive/folders/11eEFrtMI9sZWv7U6jcSKKinwycys_oKF?usp=sharing

Comment 8: As for the data augmentation for improving the performance, it might also be worth to consideration of the photoing distance. Because the current smart phones are autofocus, different photoing distances maybe lead to colors not be completely consistent. Answer) As the reviewer mentioned, distance can influence the results. To minimize the influence of the distance between smartphones and LFA kits, mobile applications provide the guideline when they take images (Please see Fig. 4a). We added a sentence as "*Since the distance between the camera of smartphone and LFA test kit might influence the image size and quality, we provided a guideline (blue line in Fig. 4a) for image capture processing.*" (line 261-263).

Comment 9: If the proposed SMART^{AI}-LFA strategy is used for clinical diagnosis to COVID-19 in POCT. How is the reproducibility for different user?

Answer) We newly added reproducibility from different users with different smartphones as *“An important parameter of the AI-assist app is universality. We confirmed the universality by validating the ability of multi-users and multi-LFA models (Fig. 4e-f). First, five smartphone users with different smartphone models (LG Q51, Galaxy A52, iPhone 12 mini, iPhone 11 Pro, and iPhone 14 Max; Supplementary Table. 4) tested the smartphone app-based diagnostics for multi-user tests under various surroundings such as indoors/outdoors, lighting conditions, and shade/sunlight with various backgrounds (Fig. 4e, Supplementary Fig. 4, and Supplementary Video 2). Every user took three images via a smartphone app from the LFA tests of nine clinical samples (6 positives for COVID-19 and 3 healthy controls), and then acquired the results. The accuracy of the 135 app-based tests with different users/smartphones was determined as 98%.”* (line 303-312 and see revised Fig.4e, Supplementary Fig. 4, and Supplementary Video 2).

Comment 10: The time from dropping sample to show assay results in smartphone should be provided.

Answer) Generally, the LFA assay time is 10-15 min. After the LFA assay is done, the time needed from taking images to returning the results is within tens of seconds (It might depend on the internet conditions. Generally within couple of seconds). We added as *“LFAs generally require assay times of up to 15 min. In app operation, the additional time needed from taking the image to returning the result is generally within tens of seconds, depending on the network and smartphones.”* (line 271-273).

Comment 11: In Eq. 1 and Eq. 2, the meaning of each parameter needs to be explained specified. Answer) We added as *“Note that N = # of train data set, y_1 = binary classification truth value of $i \rightarrow 0$ or 1 , x_1 = input data of i , θ = parameter of model, h = model, $h(x_1; \theta)$ = probability prediction of input $x_1 \rightarrow [0, 1]$ 0 to 1 value, y_1 = density truth value, \hat{y}_1 = density prediction.”* (line 501-503).

Comment 12: The one most important factors of biological assays is limit of detection (LOD). This article chose to calculate LOD based on the concentrations of nine out of ten untrained individuals/human experts correctly answered. What is the basis for defining LOD in this way?
Answer) Generally, the LOD is determined as the lowest concentration where $\geq 95\%$ (19/20) of the replicates are positive. We revised it as *“We tested the accuracy of the blind test using data sets (n = 1,500) from clinical samples. To calculate the LOD values, we prepared a second dataset (n=34: 26 positives and 8 negatives) with different concentrations, then set the LOD values under the manufacturer’s guidelines with 19/20 criteria, which represents the LOD is determined as the lowest concentration where 95% (19/20) are positive (<https://www.fda.gov/media/137302/download>).”* (lines 428-432).

Comment 13: There are several minor grammatical errors spotted throughout the reading, hence it is better to check carefully before next submission.

Answer) We corrected all the minor points, grammatical issues, and typos throughout the manuscript.

We really appreciate you again for your valuable comments, and we found the comments very helpful in clarifying the originality of our work.

Reviewer #3

General comments:

The authors identify an important opportunity for improvement of the performance of visually read lateral flow assays by using a deep learning algorithm to improve over human reader performance.

For the most part, the approaches are sound. However, the stated performance increases (97% accuracy vs 62% for untrained users) are higher than what I would expect to be possible with this method except for very low-titer samples. Based on the described methodology I am not clear to what extent the authors relied on diluted clinical vs actual low-titer clinical samples to train and validate their models. Specific comments are below on how the paper could be improved by better explaining the methodology.

Answer) We really appreciate the reviewer's helpful comment. After carefully reading the reviewer's comments, we intensively revised all the figures, introduction, discussion, and conclusion parts with additional references.

First, we revised all the figures with additional experimental data. We revised fig.1 with simplified schemes and texts to clearly show the workflow of SMART^{AI}-LFA for COVID-19 diagnostics. We deleted the wordy text to deliver the main idea. In fig. 2, We redesigned/acquired the blind test with unbiased sample distribution (discussed later); then, we moved the data of (g) the blind test, (h) Cross-reactivity, and (i) the concentration prediction to revised fig.3. In revised fig.2, we focused only the algorithm optimization. In revised fig.3, we tried to show clinical validation via blind test, cross-reactivity, concentration prediction, and daily monitoring. We deleted the blind test of NP/OP and saliva (fig. 3(b-e) and fig. 3(f-i) in the original manuscript). Instead, we showed the blind test with increased test data from n=82 to n=1500 and an evenly distributed data set for acquiring unbiased results. Moreover, we revised fig.4 with newly added results. First, we showed the schemes of the smartphone app and data flows from a smartphone to the server (AWS) with new flow charts (See fig.4a-b). After showing the ROC curves of additional clinical samples onto standard samples (See fig.4c), we compared two algorithms (SMART^{AI}-LFA and xRcovid) (See fig.4d). We newly added the universality by validating the ability of multi-users and multi-LFA models (fig.4e-

f). Then, we show the effects of training data on sensitivity and specificity (fig.4g). Finally, we demonstrated the accuracy according to test data, showing a more reliable (unbiased) performance of SMART^{AI}-LFA than humans (fig.4h).

Second, we mainly revised the introduction parts as *“Although the real-time polymerase chain reaction (RT-qPCR) test is highly sensitive, frequent on-site tests of COVID-19 using RT-qPCR are still challenging. Generally, RT-qPCR can detect viral shedding up to 17 days after the infection period, increasing unnecessary quarantine^{2,3}. Moreover, long turnaround times of RT-qPCR allow infected people to spread the viruses exponentially before getting results.”* (See lines 47-51). and we added articles with smartphone AI as *“Since the number of current smartphone users exceeds 6 billion, meaning >80% of the world's population owns smartphones (<https://www.statista.com/statistics/330695/number-of-smartphone-users-worldwide/>), the assay using a smartphone can give accessibility and affordability. Smartphone-based assays have been performed to detect sperm concentration and motility²⁷, protein biomarkers²⁸, CRISPR-read SARS-CoV-2²⁵, Zika virus²⁹, norovirus³⁰, cell migration³¹, SARS-CoV-2 variants at single-nucleotide resolution³². To further improve the performance of smartphone assays, AI-assisted assays were performed for DNA diagnosis in malaria detection²⁴, HIV rapid tests³³, CRISPR-Cas13a based SARS-CoV-2 detection²⁶, and serological SARS-CoV-2 antibody test²¹.”* (See lines 64-73).

Third, we fully redesigned the blind test starting from the unbiased data. For this, we increased the test set number from 80 to 1,500, then evenly distributed the data set to minimize the bias of test results. We trained 8,005 images (positive: 5,667 and negative 2,338) and tested 1,500 images (positive: 1,000 and negative 500). All the blind test images are prepared from the COVID-19 patient/healthy control samples. We collected all the clinical samples from Seoul St. Mary's Hospital with Institutional Review Board Committee approval (KC21TIDI0134K) and acquired all the Ct values from the patient samples. The Ct value is not exactly inversely proportional to the color intensity of LFA test results; therefore, we prepared the blind tests set using the color chart under the LFA manufacturer's guidelines. Generally, the color chart consisted of 10 classifications; then, we evenly prepared the blind test images (high 250 images, mid 250 images, low 250 images, and very low 250 images) to prevent bias. We revised as follows: *“We determined the accuracy of the blind test using 1,500 test images from untrained individuals (n=10), human experts (n=10), and SMART^{AI}-LFA (Fig. 1b). Clinical samples*

($n=65$, COVID-19 patients: 45 and healthy controls: 20) were tested to validate the clinical predictability of SMART^{AI}-LFA. We arranged the training, test and validation datasets for algorithm development (**Supplementary Table 1-2, Supplementary Fig. 1**).” (line 117-121) and “We assess the blind test using three different groups: untrained individuals, human experts, and SMART^{AI}-LFA using 1,500 test images (1,000 positives and 500 negatives). We collected clinical samples of NP/OP samples from COVID-19 patients at Seoul St. Mary’s Hospital. The information pertained to SARS-CoV-2 patients ($n=45$) and healthy controls ($n=20$) including sample collection, variants, sex, age, and Ct value (Supplementary Table. 3). All the samples were analyzed with RT-qPCR, then conducted the LFA assay. Finally, the positive data from LFA assay were classified into four groups, i.e., high/middle/middle-low/low titer, with the aid of the color chart level (high with levels 10–8, middle with level 7–5, middle-low with level 4–3, and low titer with level 2–1 for positive, and negative with level 0). For positive data ($n=1,000$), we evenly distributed data across four groups ($n=250$). We prepared negative data (health controls, $n=500$). For the blind test, ten untrained individuals and ten human experts tested each of the 150 test images, which included 25 high, 25 middle, 25 middle-low, 25 low, and 50 negative data. The sum of the blind tests for both untrained individuals and human experts was 1500 images.

The ROC curve shows a general overview of the three different models; a larger value of the area under the curve (AUC) indicates a better classifier. From the ROC curve in Fig. 3b, we observe larger AUCs for SMART^{AI}-LFA (1.00) compared with that for untrained individuals (0.79) and human experts (0.86); this demonstrates that SMART^{AI}-LFA is an excellent classifier for clinical assays. Fig. 3c shows the table for the three different groups, which indicates a considerable enhancement in sensitivity and specificity using a SMART^{AI}-LFA (100 and 100%) compared with untrained individuals (72.9 and 86.0%) and human experts (83 and 88.2%). To closely study the reason for the accuracy reaching 100%, we explored the effect of the training and test datasets discussed later (Fig. 4g-h).

We present three positive clinical sample images in Fig. 3d to clarify the AI’s decision ability. Fig. 3e shows the evidence of a positive test line of Fig. 3d with contrast enhancement. Although all groups can provide the correct answer as seen from the first image, only SMART^{AI}-LFA can predict the positive samples from the third image, which confirms that deep learning-assisted SMART^{AI}-LFA can provide precision decisions superior to those of human experts.” (line 187-215)

Fourth, we relocated clinical validation of cross-reactivity, concentration prediction, and daily monitoring in fig. 3. (See revised fig.3 and manuscript parts).

Fifth, we fully revised fig. 4. We added *Supplementary Videos* to show the SMART^{AI}-LFA's operation. To depict the data flows between smartphones and servers (Fig. 4b), we revised as "**Fig. 4b** shows a schematic diagram depicting the data flows between smartphones and servers. The image of the test results acquired from a smartphone are sent to a server (Amazon Web Service, AWS) with image bytes, and the test results determined by the deep learning algorithm are transmitted back in the JSON format to the smartphone applications. We imported the Flask module, a micro web framework written in Python, of SMART^{AI}-LFA and created a Flask web server on Amazon Cloud. CNN models with YOLOv3 and ResNet-18 are used to realize the sample-to-answer platform of the COVID-19 POCT. LFAs generally require assay times of up to 15 min. In app operation, the additional time needed from taking the image to returning the result is generally within tens of seconds, depending on the network and smartphones." (line 265-273). To check the AI-assist app's universality, we first validated the ability of multi-users with multiple smartphones (Fig. 4e-f) as "An important parameter of the AI-assist app is universality. We confirmed the universality by validating the ability of multi-users and multi-LFA models (**Fig. 4e-f**). First, five smartphone users with different smartphone models (LG Q51, Galaxy A52, iPhone 12 mini, iPhone 11 Pro, and iPhone 14 Max; **Supplementary Table. 4**) tested the smartphone app-based diagnostics for multi-user tests under various surroundings such as indoors/outdoors, lighting conditions, and shade/sunlight with various backgrounds (**Fig. 4e, Supplementary Fig. 4, and Supplementary Video 2**). Every user took three images via a smartphone app from the LFA tests of nine clinical samples (6 positives for COVID-19 and 3 healthy controls), and then acquired the results. The accuracy of the 135 app-based tests with different users/smartphones was determined as 98%." (line 303-312). Then, we validated the universality by testing an additional seven different LFA models (Fig. 4f). "Second, we validated the universality by testing an additional seven different LFA models (**Fig. 4f, Supplementary Fig. 5, Supplementary Table. 5, and Supplementary Video 3**). We used Panbio COVID-19 Ag, BIO CREDIT COVID-19 Ag, SGT-flex COVID-19 Ag, GENEDIA COVID-19, Humasis COVID-19 Ag Test, Genbody COVID-19 Ag, InstaView COVID-19. Note that we carried out no additional training, meaning that we used the SMART^{AI}-LFA algorithm trained with LFA model 1 (COVID-19 Ag LFA kits, Calth Inc.). Then, we validated SMART^{AI}-LFA using three LFA models (n=360, Rapigen, SD biosensor, and

Yuhan, Republic of Korea, See **Supplementary Table. 5**) and finally tested seven different models with smartphone app-based image acquisition ($n=9,450$). Every LFA kit was tested with 1,350 data (900 positives and 450 negatives). To avoid overfitting during the learning process, we tried to validate the model using the different LFA models from different manufacturers used in learning and testing. The sensitivity and specificity of Model 1 (Calth Inc.) from the confusion matrix are 99.6 % and 99.3%, respectively. Interestingly, the average sensitivity and specificity with seven different LFA models from 9,450 app-based tests were 94% and 89.7%, respectively. The total averaged sensitivity and specificity from eight different LFA models (LFA model 1 and 7 different models) were determined as 94.8% and 90.9%, respectively, indicating good universality of SMART^{AI}-LFA.” (line 313-328). In fig. 4g, we tried to determine whether the training data is associated with the tunability of sensitivity and specificity as “The aim of **Fig. 4g** has been to identify the tunability of sensitivity and specificity. Generally, LFA manufacturers control the sensitivity and specificity of commercial LFA by optimizing chemistry, materials, and LFA design. To determine whether the training data is associated with the tunability of sensitivity and specificity, we controlled the ratio of the low titers in the training data. We trained the algorithm with the existing dataset (SMART^{AI}-LFA) and validated ($n=360$, Rapigen, SD biosensor, and Yuhan, Republic of Korea, See **Supplementary Table. 5**) and finally tested 10,908 data (7,326 positives and 3,582 negatives). We visualized the confusion matrix, which allows the extraction of the true negatives (TN), true positives (TP), false negatives (FN), and false positives (FP). The true positive and true negative portions of the confusion matrix indicate the sensitivity and specificity of the clinical assay, respectively. With more low titer (high/middle/middle-low/low titer=20/25/30/25%) of test data, we acquired higher sensitivity for eight different LFA models (LFA model 1 and 7 different models) as 94.8% with 90.9% specificity. Meanwhile, the algorithm trained without low titer samples (high/middle/middle-low/low titer=27/33/40/0 %) yielded higher specificity (99 %) with less sensitivity (86%).”. In fig. 4h, we tried to show the effects of test data on the accuracy. “Finally, we showed the accuracy according to test data, including three different low-titer ratios (31, 61, and 91%) (**Fig. 4h**). We designed blind tests with 150 data (100 positives and 50 negatives) for each untrained individual ($n=3$) and human expert ($n=3$), then compared it with SMART^{AI}-LFA. We evenly distributed the tests (high/middle/middle-low) with low-titer ratios. For example, for a 31% low titer design, we prepared 23% high, 23% middle, 23% middle-low, and 31% low. Thus, we investigated the significant decrease in the accuracy of humans for lower concentrations, representing the human bias under lower concentrations. With an increase in the lower titer ratio (31 to 91%), we observed a significant

decrease in the accuracy of untrained individuals (72.6 to 51.6%) and human experts (80.2 to 57.6%). Meanwhile, we noticed that the accuracy of SMART^{AI}-LFA is maintained at over 99% (100 to 99.3%). The accuracy data in Fig. 4h explicitly confirms the more reliable performance of SMART^{AI}-LFA in comparison to humans (untrained individuals and human experts).” (line 330-356).

Lastly, we added major limitation of SMART^{AI}-LFA at the end of “Results and Discussions” part as *“The SMART^{AI}-LFA study had a major limitation: image quality depends on the smartphone. the high-end smartphone has an automatic filter function, which decreases data accuracy because the smartphone automatically corrects its image quality. Therefore, further studies need to focus on acquiring raw data. Another limitation is that the accuracy could be reduced if the test surroundings are outside the training data. Testing the surrounding out-of-distribution training data potentially limits the accuracy of the current deep learning model. Retest signs in the image acquisition stage can be considered a solution to address these problems. Guided by the retest sign, images can be obtained within the training-data distribution.”* (line 357-364). Then, we added new sentences in the conclusion parts as *“(3) Validation of multi-users and multi-LFA models showed the universality of SMART^{AI}-LFA. Since smartphone users exceed 6 billion, SMART^{AI}-LFA with smartphone can improve the POCT along with affordability.”* (line 376-378)

We hope the manuscript, with careful revisions, meets your high standards. Below we provide the point-by-point responses. All modifications in the manuscript have been highlighted in red.

Comment 1: Line 27: "Current self-testing for COVID-19 detection suffers from low accuracy and reading ambiguities attributable to highly subjective readings performed by untrained individuals." I don't think this is a primary issue. Real sensitivity of antigen LFAs lags PCR of course, but that is primarily a function of the assay chemistry, not of the reading methodology. Answer) I totally agree with the reviewer's comments. The performance and its accuracy are mainly dependent on the assay chemistry. To increase the LFA performance, we proposed “PCR-like Performance of Rapid Test with Permselective Tunable Nanotrap” and are now in the revision process of Nat Comm. To meet the reviewer's suggestion, we toned down the sentence from *"Current self-testing for COVID19 detection suffers from low accuracy and*

reading ambiguities attributable to highly subjective readings performed by untrained individuals." to "current self-testing for COVID-19 detection suffers from low accuracy, mainly due to the LFA sensitivity and reading ambiguities." (line 27-28).

Comment 2: Line 47: "Although the real-time polymerase chain reaction (RT-PCR) test is highly sensitive, one-time monitoring can detect viral shedding up to 17 days after the infection period": what does one-time monitoring refer to? With PCR? And is that small amount of shedding relevant for transmission?

Answer) We changed confusion expression in introduction parts, from "*Although the real-time polymerase chain reaction (RT-PCR) test is highly sensitive, one-time monitoring can detect viral shedding up to 17 days after the infection period³. Therefore, rapid testing for COVID-19 is essential for minimizing virus transmission.*" to "*Although the real-time polymerase chain reaction (RT-qPCR) test is highly sensitive, frequent on-site tests of COVID-19 using RT-qPCR are still challenging. Generally, RT-qPCR can detect viral shedding up to 17 days after the infection period, increasing unnecessary quarantine^{2,3}. Moreover, long turnaround times of RT-qPCR allow infected people to spread the viruses exponentially before getting results.*" (See lines 47-51).

Comment 3: Line 76 - provide reference for REASSURED.

Answer) We revised as "*Further, a smartphone-based AI provides a great opportunity to meet the REASSURED criteria (Real-time connectivity, Ease of specimen collection, Affordable, Sensitive, Specific, User friendly, Rapid and robust, Equipment free and Deliverable to end-users)²², which are the new criteria for digital connectivity.*" (line 94-97).

Comment 4: Line 92. Describe "augmented dataset". Explain why it does not introduce bias and if and why it improves the accuracy of the deep learning method.

Answer) When the starting data is biased, as the reviewer mentioned, the augmented data will be biased. We newly added a sentence as "*To avoid bias due to data augmentation, we prepared images under various surroundings; in turn, we could increase the accuracy with data augmentation (Fig. 2e-f).*" (line 170-172).

Comment 5: Line 94; "After qualifying the clinical samples, we trained the model with additional clinical samples (n = 8,005) and acquired 99.2% accuracy when using the smartphone application." How does this compare to visually read accuracy with the same data set? Also, the NAVICA app (note spelling!) is not an AI reader, it just captures images. I do not understand the relevance of the sentence starting in line 96.

Answer) First, we deleted "*Abbott developed the NOVICA application to pair with their rapid antigen test for COVID 19. Untrained individuals use the NOVICA app to obtain the test results; However, the decision is made by human experts, not by AI.*" in revised manuscript. Second, to address 'visually read accuracy', we fully redesigned the blind test starting from the unbiased data. For this, we increased the test set number from 80 to 1,500, then evenly distributed the data set to minimize the bias of test results. (See line 117-121, line 187-215, fig. Fig. 3. (a-e) Clinical validation via blind tests). we observe larger AUCs for SMART^{AI}-LFA (1.00) compared with that for untrained individuals (0.79) and human experts (0.86); this demonstrates that SMART^{AI}-LFA is an excellent classifier for clinical assays.

Moreover, in revised fig. 4h, we observed a significant decrease in the accuracy of untrained individuals (72.6 to 51.6%) and human experts (80.2 to 57.6%) with an increase in the lower titer ratio (31 to 91%) of test data. Meanwhile, we noticed that the accuracy of SMART^{AI}-LFA is maintained at over 99.0%. The accuracy data in Fig. 4h explicitly confirms the more reliable (less biased) performance of SMART^{AI}-LFA in comparison to humans (untrained individuals and human experts) (See line 345-356 and fig. 4h). Note that we used same data set for untrained individuals, human experts, and SMART^{AI}-LFA, respectively.

Comment 6: Line 100 - which commercial LFA?

Answer) We revised from "*We design an app to pair with the commercial COVID-19 Ag LFA for antigen diagnostic tests.*" to "*We first designed an app to pair with AllCheck COVID19 Ag (Calth Inc.), then tested the seven commercial COVID-19 models such as Panbio COVID-19 Ag (Abbott), BIO CREDIT COVID-19 Ag (Rapigen), SGT-flex COVID-19 Ag (Sugentech), GENEDIA COVID-19 (GCMS), COVID-19 Ag Test (Humasis), COVID-19 Ag (Genbody), and InstaView COVID-19 (SGmedical).*" (line 109-113).

Comment 7: Line 108 - "untrained individuals, human experts, and SMART^{AI}-LFA to be 62.7, 76.6, and 97.9%" This enormous difference between trained and untrained users, and between trained users and the AI seems extremely hard to believe UNLESS the patient samples were strongly biased towards very low viral loads. Most patients test during the acute phase of the infection and a positive antigen line is usually clear as day. The main benefit of AI-supported reading would likely be for very weak lines, corresponding to low viral loads prior to the onset of symptoms or as the infection is waning, but that is not when most untrained users are performing home tests.

Answer) As mentioned in the answer of comments 1 and 5, we tried to show the unbiased blind test. Importantly, in *revised fig. 4h*, we showed the accuracy of SMART^{AI}-LFA is maintained at over 99.0% while accuracy of untrained individuals (72.6 to 51.6%) and human experts (80.2 to 57.6%) rapidly dropped with an increase in the lower titer ratio (31 to 91%) of test data. The *fig. 4h* data explicitly confirms the less biased performance of SMART^{AI}-LFA in comparison to humans. (Please see the fully revised manuscript for more details).

We totally agreed with the comments, *'Most patients test during the acute phase of the infection, and a positive antigen line is usually clear as day.'* However, recently published papers in NEJM (New Engl J Med 383, e120 (2020) and BMJ (Bmj 372, n208 (2021)) claimed new diagnostic strategies for handling pandemics. Authors reported that the best Covid filter could be achieved using frequent, low-cost, simple, and rapid tests because SARS-CoV-2 quickly grows and spreads out exponentially. Therefore, rapid and frequent testing for COVID-19 is essential for minimizing virus transmission, especially before recognizing symptoms and in asymptomatic cases. We expect smartphone AI can help the strategy of NEJM and BMJ in the near future since it provides more accuracy and affordability with REASSURED criteria. To address reviewer's concerns, we revised as *"Since the number of current smartphone users exceeds 6 billion, meaning >80% of the world's population owns smartphones* (<https://www.statista.com/statistics/330695/number-of-smartphone-users-worldwide/>), *the assay using a smartphone can give accessibility and affordability. Smartphone-based assays have been performed to detect sperm concentration and motility²⁷, protein biomarkers²⁸, CRISPR-read SARS-CoV-2²⁵, Zika virus²⁹, norovirus³⁰, cell migration³¹, SARS-CoV-2 variants at single-nucleotide resolution³². To further improve the performance of smartphone assays, AI-assisted assays were performed for DNA diagnosis in malaria detection²⁴, HIV rapid tests³³, CRISPR-Cas13a based SARS-CoV-2 detection²⁶, and serological SARS-CoV-2*

antibody test21." (line 64-73).

Comment 8: Line 158 please see my earlier comment on data augmentation.

Answer) We newly added sentence as *"To avoid bias due to data augmentation, we prepared images under various surroundings; in turn, we could increase the accuracy with data augmentation (Fig. 2e-f).*" (line 170-172).

Comment 9: Line 213: Dilution with PBS to create low-titer samples is problematic as agents that typically cause interference and non-specific binding, especially at levels mimicking a weak positive line would also be diluted out. So results with diluted samples, for a given viral titer, will look more accurate than they would be in real undiluted clinical samples.

Answer) Thanks for your exact comments. As the reviewer mentioned, PBS dilution reduces interference and non-specific binding. However, in the commercialized COVID-19 assay protocol of LFA, one usually uses extraction buffer with a swab sample (10:1). Since the LFA itself used the diluted samples, we expect the PBS dilution is not the critical factor for validation of AI. Moreover, since we tested untrained individuals, human experts, and AI with the same data sets, the PBS dilution might not affect the performance validations between humans and AI.

Comment 10: Line 214 - were the 83% with intermediate and low viral loads undiluted clinical samples or diluted clinical samples?

Answer) We deleted *"The clinical tests are validated using a blinded test (Fig. 3a). To strengthen this model, we start with algorithm, which is trained with standard samples (n=8,914), and then, we retrain the algorithm adding clinical samples (n = 8,005) from COVID 19 patients (n = 45) and healthy controls (n = 20); we call this algorithm SMART^{AI}-LFA (clinical). We first prepared COVID 19 positive samples with different virus titers to achieve a higher accuracy for a lower virus titer. Then, we diluted the clinical samples with 1×phosphatebuffered saline (PBS) for training, and prepared samples with different virus titers. Finally, we included 83% with intermediate (26 < Ct < 30) and low viral loads (Ct ≥30) to demonstrate the ability of performance enhancement, and we acquired the digital images.*

Then, we trained the negative controls from the healthy controls. Like positive control, we prepared several negative control samples, which include healthy control, diluted samples from healthy controls, and PBS only samples. Then, we trained an invalid test. The main invalid case in the LFA test was attributed to the absence of a control line caused by the fabrication failure of the LFA kit. We deliberately prepared an invalid case by fabricating an LFA kit with no control line and then trained the invalid kit using the deep learning algorithm. Therefore, we trained all cases, i.e., COVID 19 positive (n = 11,468), negative (n = 5,451), and invalid (n = 3,076) from the training data.", Then, added newly designed blind tests as “We assess the blind test using three different groups: untrained individuals, human experts, and SMART^{AI}-LFA using 1,500 test images (1,000 positives and 500 negatives). We collected clinical samples from COVID-19 patients at Seoul St. Mary’s Hospital. The information pertained to SARS-CoV-2 patients (n=45) and healthy controls (n=20) including sample collection, variants, sex, ages, and Ct values (**Supplementary Table. 3**). All the samples were analyzed with RT-qPCR, then conducted the LFA assay. Finally, the positive data from LFA assay were classified into four groups, i.e., high/middle/middle-low/low titer, with the aid of the color chart level (high with levels 10–8, middle with level 7–5, middle-low with level 4–3, and low titer with level 2– 1 for positive, and negative with level 0). For positive data (n=1,000), we evenly distributed data across four groups (n=250). We prepared negative data (health controls, n=500). For the blind test, ten untrained individuals and ten human experts tested each of the 150 test images, which included 25 high, 25 middle, 25 middle-low, 25 low, and 50 negative data. The sum of the blind tests for both untrained individuals and human experts was 1500 images.

The ROC curve shows a general overview of the three different models; a larger value of the area under the curve (AUC) indicates a better classifier. From the ROC curve in **Fig. 3b**, we observe larger AUCs for SMART^{AI}-LFA (1.00) compared with that for untrained individuals (0.79) and human experts (0.86); this demonstrates that SMART^{AI}-LFA is an excellent classifier for clinical assays. **Fig. 3c** shows the table for the three different groups, which indicates a considerable enhancement in sensitivity and specificity using a SMART^{AI}-LFA (100 and 100%) compared with untrained individuals (72.9 and 86.0%) and human experts (83 and 88.2%). To closely study the reason for the accuracy reaching 100%, we explored the effect of the training and test datasets discussed later (**Fig. 4g-h**).

We present three positive clinical sample images in **Fig. 3d** to clarify the AI’s decision ability. **Fig. 3e** shows the evidence of a positive test line of **Fig. 3d** with contrast enhancement. Although all groups can provide the correct answer as seen from the first image, only SMART^{AI}-LFA can predict the positive samples from the third image, which confirms that deep

learning-assisted SMART^{AI}-LFA can provide precision decisions superior to those of human experts.” (line 187-215).

Comment 11: line 462 - data augmentation - it would appear that there are still possible biases from the selection of the types of variations in ambient light and imaging conditions that were selected that would create apparent improvements in accuracy that could not then be generalized widely for real use.

Answer) When the starting data is biased, as the reviewer mentioned, the augmented data will be biased. We newly added a sentence as *“To avoid bias due to data augmentation, we prepared images under various surroundings; in turn, we could increase the accuracy with data augmentation (Fig. 2e-f).” (line 170-172).*

Distance between smartphones and LFA kits can also influence the imaging conditions. To minimize the influence of the distance, mobile applications provide the guideline when they take images (Please see Fig. 4a). We added a sentence as *“ Since the distance between the camera of smartphone and LFA test kit might influence the image size and quality, we provided a guideline (blue line in Fig. 4a) for image capture processing. ” (line 261-263).*

Lastly, we added major limitation of SMART^{AI}-LFA at the end of “Results and Discussions” part as *“Another limitation is that the accuracy could be reduced if the test surroundings are outside the training data. Testing the surrounding out-of-distribution training data potentially limits the accuracy of the current deep learning model. Retest signs in the image acquisition stage can be considered a solution to address these problems. Guided by the retest sign, images can be obtained within the training-data distribution.” (line 360-364).*

Our newly added data in fig.4 were mainly tested using smartphones under various conditions (*please see the three supplementary videos*), and we hope the acquired performance could address the reviewer’s concerns.

We really appreciate you again for your valuable comments, and we found the comments very helpful in clarifying the originality of our work. We hope the manuscript, with careful revisions, meets your high standards.

REVIEWERS' COMMENTS

Reviewer #1 (Remarks to the Author):

"I acknowledge the substantial revision of the manuscript and have no further comments. However, I suggest the authors include an IRB statement in the manuscript before publication."

Reviewer #2 (Remarks to the Author):

The author has responded to my comments, but there are still some questions that are not answered positively or disagree with my views, and these aspects are reflected in the innovation and importance of the work. I believe that some discussion is still needed before the manuscript can be published, with the following comments

Comments :

1. Lateral flow assay (LFA) is a qualitative coarse measurement with low sensitivity but rapid and user-friendly. When the concentration of antigen is high enough, the color of the red line in LFA is so obvious that it can be directly observed by the human eye. Artificial Intelligence (AI) by smartphone for LFA is superfluous in this case. The significance of this work should lie in how LFA results that cannot be recognised by the human eye can be identified through mobile phone AI. I don't think the author gives a correct description of importance and innovation.
2. I think the method would be very interesting if the sensitivity of LFA detection could be enhanced by mobile phone AI or if antigens could be quantified. Whether the work could be improved in this respect by the authors?
3. The amount of antigen in a person's body is gradually raised after infection with the SARS-CoV-2. This means that in the early stages of infection, antigen levels are extremely low and almost undetectable by LFA. Can early infections be detected through LFA with smartphone AI? How much earlier can SARS-CoV-2 infections be detected than conventional detection methods? Is it possible to detect early SARS-CoV-2 infection? How does it compare with conventional methods? How soon can the infection be detected?

4. Limit of detection (LOD) should be the lowest concentration of the antigen tested by LFA with smartphone AI. I don't think the author's expression is accurate and easily confused.

5. It is worthwhile for the authors to explore whether there are other more important features of the combination of smartphone and LFA. The advantages that mobile phone AI embodies in that work and the innovations made by the authors are moderate, but it is certainly a good attempt.

Reviewer #3 (Remarks to the Author):

Thank you for your detailed responses to all the reviewer's comments. I believe they were addressed appropriately.

Response letter

Reviewer #2

Comment 1: Lateral flow assay (LFA) is a qualitative coarse measurement with low sensitivity but rapid and user-friendly. When the concentration of antigen is high enough, the color of the red line in LFA is so obvious that it can be directly observed by the human eye. Artificial Intelligence (AI) by smartphone for LFA is superfluous in this case. The significance of this work should lie in how LFA results that cannot be recognised by the human eye can be identified through mobile phone AI. I don't think the author gives a correct description of importance and innovation.

Answer) I appreciate your comment regarding the importance and innovation of our work. We revised discussion parts as *“The availability of multi-user and multi-model systems with a reduced bias for low-titer samples provides a promising avenue for achieving breakthroughs in the early detection of infections using SMART^{AI}-LFA technology. By circumventing the reading ambiguities encountered in LFA tests, we anticipate that SMART^{AI}-LFA will be particularly effective in detecting infections in their early stages.”* (line 388~392)

Comment 2: I think the method would be very interesting if the sensitivity of LFA detection could be enhanced by mobile phone AI or if antigens could be quantified. Whether the work could be improved in this respect by the authors?

Answer) We appreciate the reviewer's comment on the potential to improve the sensitivity of LFA detection and quantify antigens using mobile phone AI. In this study, we developed a positive/negative algorithm that requires only a small amount of computation and performed experiments to validate its efficacy. However, developing quantification algorithms for antigen-antibody responses presents several technical challenges that must be overcome before they can be ported to AWS. Such algorithms are often computationally intensive and require significant resources. Therefore, we agree with the reviewer's suggestion and intend to pursue future work to develop a new algorithm for quantification that can more accurately measure the antigen and antibody reactions in clinical samples.

Comment 3: The amount of antigen in a person's body is gradually raised after infection with the SARS-CoV-2. This means that in the early stages of infection, antigen levels are extremely

low and almost undetectable by LFA. Can early infections be detected through LFA with smartphone AI? How much earlier can SARS-CoV-2 infections be detected than conventional detection methods? Is it possible to detect early SARS-CoV-2 infection? How does it compare with conventional methods? How soon can the infection be detected?

Answer) We have demonstrated, as shown in Figures 3i and 4h, that the diagnostic performance of the SMART^{AI}-LFA is enhanced at a low viral load. However, due to the large coefficient of variation (CV) produced by LFA, the accuracy and LOD of SMART^{AI}-LFA may not match the performance of PCR. Nonetheless, the use of SMART^{AI}-LFA provides a powerful, highly sensitive, and accessible diagnostic method that can be used on-site for infectious diseases. As mentioned in answer #1, we reflected it as “*The availability of multi-user and multi-model systems with a reduced bias for low-titer samples provides a promising avenue for achieving breakthroughs in the early detection of infections using SMART^{AI}-LFA technology. By circumventing the reading ambiguities encountered in LFA tests, we anticipate that SMART^{AI}-LFA will be particularly effective in detecting infections in their early stages.*” (line 388~392)

Comment 4: Limit of detection (LOD) should be the lowest concentration of the antigen tested by LFA with smartphone AI. I don't think the author's expression is accurate and easily confused.

Answer) Generally, the LOD is determined as the lowest concentration where $\geq 95\%$ (19/20) of the replicates are positive. We revised it as “*We evaluated the accuracy of the blind test using a dataset of clinical samples (n=1,500). To determine the limit of detection (LOD), we prepared a second dataset (n=34: 26 positives and 8 negatives) with varying concentrations. We then determined the LOD values using the manufacturer's guidelines, which indicate that the LOD is the lowest concentration at which 95% (19/20) of the tests are positive (as outlined in <https://www.fda.gov/media/137302/download>).*” (lines 438-443).

Comment 5: It is worthwhile for the authors to explore whether there are other more important features of the combination of smartphone and LFA. The advantages that mobile phone AI embodies in that work and the innovations made by the authors are moderate, but it is certainly a good attempt.

Answer) Thank you for your comment and feedback on our work. We appreciate your suggestion to explore other potentially more important features of the combination of smartphone and LFA. We will continue to investigate the potential of the smartphone-LFA combination in our future work and strive to identify new and innovative applications that can improve disease diagnosis and management. We thank you again for your valuable feedback, which will guide our future research.